# Fisher Efficient Inference of Intractable Models

**Song Liu**
University of Bristol
The Alan Turing Institute, UK
song.liu@bristol.ac.uk

**Takafumi Kanamori**
Tokyo Institute of Technology,
RIKEN, Japan
kanamori@c.titech.ac.jp

**Wittawat Jitkrittum**
Max Planck Institute
for Intelligent Systems, Germany
wittawat@tuebingen.mpg.de

**Yu Chen**
University of Bristol, UK
yc14600@bristol.ac.uk

## Abstract

Maximum Likelihood Estimators (MLE) has many good properties. For example, the asymptotic variance of MLE solution attains equality of the asymptotic Cramér-Rao lower bound (efficiency bound), which is the minimum possible variance for an unbiased estimator. However, obtaining such MLE solution requires calculating the likelihood function which may not be tractable due to the normalization term of the density model. In this paper, we derive a Discriminative Likelihood Estimator (DLE) from the Kullback-Leibler divergence minimization criterion implemented via density ratio estimation and a Stein operator. We study the problem of model inference using DLE. We prove its consistency and show that the asymptotic variance of its solution can attain the equality of the efficiency bound under mild regularity conditions. We also propose a dual formulation of DLE which can be easily optimized. Numerical studies validate our asymptotic theorems and we give an example where DLE successfully estimates an intractable model constructed using a pre-trained deep neural network.

## 1   Introduction

Maximum Likelihood Estimation (MLE) has been a classic choice of density parameter estimator. It can be derived from the Kullback-Leibler (KL) divergence minimization criterion and the resulting algorithm simply maximizes the likelihood function (log-density function) over a set of observations. The solution of MLE has many attractive asymptotic properties: the asymptotic variance of MLE solutions reach an asymptotic lower bound of all unbiased estimators [5, 24].

However, learning via MLE requires evaluating the normalization term of the density function; it may be challenging to apply MLE to learn a complex model that has a computationally intractable normalization term. A partial solution to this problem is approximating the normalization term or the gradient of the likelihood function numerically. Many methods along this line of research have been actively studied: importance-sampling MLE [25], contrastive divergence [12] and more recently amortized MLE [33]. While the computation of the normalization term is mitigated, these sampling-based approximate methods come at the expense of extra computational burden and estimation errors.

The issue of intractable normalization terms has led to the develoment of other approaches other than the KL divergence minimization. For example, Score Matching (SM) [13] minimizes the Fisher divergence [26] between the data distribution and a model distribution which is specified by the gradient (with respect to the input variable) of its log density function. Its computation does not

require the evaluation of the normalization term, thus SM does not suffer from the intractability issue. Extensions of SM has been used for infinite dimensional exponential family models [28], non-negative models [14, 35] and high dimensional graphical models fitting [17].

Other than the Fisher divergence, a kernel-based divergence measure known as Kernel Stein Discrepancy (KSD) [4, 19] has been proposed as a test statistic for goodness-of-fit testing to measure the difference between a data and a model distribution, without the hassle of evaluating the normalization term. It reformulates the kernel Maximum Mean Discrepancy (MMD) [9] with a Stein operator [29, 8, 23] which is also defined using the gradient of the log density function. For the same reason as in SM, the KSD can be estimated when applied to a density model with an intractable normalizer. The last few years have seen many applications of KSD such as variational inference [18], sampling [23, 3], and score function estimation [16, 27] among others. KSD minimization is a natural candidate criterion for fitting intractable models [2]. However, the divergence measure defined by the KSD is directly characterized by the kernel used. Unlike in the case of goodness-of-fit testing where the kernel may be chosen by maximizing the test power [15], to date, there is no clear objective for choosing the right kernel in the case of model fitting.

By contrast, KL divergence has been a classic discrepancy measure for model fitting. The question that we address is: can we construct a generic model inference method by minimizing the KL divergence without the knowledge of the normalization term? In this paper, we present a novel *unnormalized* model inference method, Discriminative Likelihood Estimation (DLE), by following the KL divergence minimization criterion. The algorithm uses a technique called Density Ratio Estimation [31] which is conventionally used to estimate the ratio between two density functions from two sets of samples. We adapt this method to estimate the ratio between a data and an unnormalized density model with the help of a Stein operator. We then use the estimated ratio to construct a surrogate to KL divergence which is later minimized to fit the parameters of an *unnormalized* density function. The resulting algorithm is a $\min\max$ problem, which we show can be conveniently converted into a min-min problem using Lagrangian duality. No extra sampling steps are required.

We further prove the consistency and asymptotic properties of DLE under mild conditions. One of our major contributions is that we prove the proposed estimator can also attain the asymptotic Cramér-Rao bound. Numerical experiments validate our theories and we show DLE indeed performs well under realistic settings.

## 2 Background

### 2.1 Problem: Intractable Model Inference via KL Divergence Minimization

Consider the problem of estimating the parameter $\boldsymbol{\theta}$ of a probability density model $p(\boldsymbol{x}; \boldsymbol{\theta})$ from a set of i.i.d. samples: $X_q := \{\boldsymbol{x}_q^{(i)}\}_{i=1}^{n_q} \overset{\text{i.i.d.}}{\sim} Q$ where $Q$ is a probability distribution whose density function is $q(\boldsymbol{x})$. One idea is minimizing the sample approximated KL divergence from $p_{\boldsymbol{\theta}}$ to $q$:

$$\min_{\boldsymbol{\theta}} \text{KL}\left[q|p_{\boldsymbol{\theta}}\right] = \min_{\boldsymbol{\theta}} \mathbb{E}_q \left[\log \frac{q(\boldsymbol{x})}{p(\boldsymbol{x}; \boldsymbol{\theta})}\right] = C - \max_{\boldsymbol{\theta}} \mathbb{E}_q \left[\log p(\boldsymbol{x}; \boldsymbol{\theta})\right]$$

$$\approx C - \max_{\boldsymbol{\theta}} \frac{1}{n_q} \sum_{i=1}^{n_q} \log p(\boldsymbol{x}_q^{(i)}; \boldsymbol{\theta}),$$

where $C$ is a constant that does not depend on $\boldsymbol{\theta}$. The last line uses $X_q$ to approximate the expectation over $q(\boldsymbol{x})$. This technique is known as Maximum Likelihood Estimation (MLE).

Despite many advantages, MLE is unfit for intractable model inference. Consider for instance a density model $p(\boldsymbol{x}; \boldsymbol{\theta}) := \frac{\bar{p}(\boldsymbol{x}; \boldsymbol{\theta})}{z(\boldsymbol{\theta})}$, where $\bar{p}(\boldsymbol{x}; \boldsymbol{\theta})$ is a positive multilayer neural network parametrized by $\boldsymbol{\theta}$, $Z(\boldsymbol{\theta}) = \int \bar{p}(\boldsymbol{x}; \boldsymbol{\theta}) \mathrm{d}\boldsymbol{x}$ is the normalization term which guarantees that $p(\boldsymbol{x}; \boldsymbol{\theta})$ integrates to 1 over its domain. In this example, $Z(\boldsymbol{\theta})$ does not have a computationally tractable form; therefore, MLE cannot be used without approximating the likelihood function or its gradient using numerical methods such as Markov chain Monte Carlo (MCMC).

However, there is an *alternative approach* to minimizing the KL divergence: $\text{KL}\left[q|p_{\boldsymbol{\theta}}\right]$ is an expectation of the log-ratio $\log \frac{q(\boldsymbol{x})}{p(\boldsymbol{x}; \boldsymbol{\theta})}$ with respect to the data distribution $q(\boldsymbol{x})$. If we have access to $\frac{q(\boldsymbol{x})}{p(\boldsymbol{x}; \boldsymbol{\theta})}$, we can approximate this KL by taking the average of the density ratio function over samples $X_q$, and

the density model parameter $\boldsymbol{\theta}$ can be subsequently estimated by minimizing this approximation to the KL divergence.

## 2.2 Two Sample Density Ratio Estimation

Traditionally, Density Ratio Estimation (DRE) [30, 31] refers to estimating the ratio of two unknown densities from their samples. Given *two* sets of i.i.d. samples drawn separately from distributions $Q$ and $P$: $X_q := \{\boldsymbol{x}_q^{(i)}\}_{i=1}^{n_q} \sim Q, X_p := \{\boldsymbol{x}_p^{(i)}\}_{i=1}^{n_p} \sim P, \boldsymbol{x}_q, \boldsymbol{x}_p \in \mathbb{R}^d$, where distribution $Q$ and $P$ have density functions $q(\boldsymbol{x})$ and $p(\boldsymbol{x})$ respectively. We hope to estimate the ratio $\frac{q(\boldsymbol{x})}{p(\boldsymbol{x})}$.

We can model the density ratio using a function $r(\boldsymbol{x}; \boldsymbol{\delta})$ parameterized by $\boldsymbol{\delta}$. To obtain the parameter $\boldsymbol{\delta}$, we minimize the KL divergence $\mathrm{KL}[q|q_{\boldsymbol{\delta}}]$ where $q(\boldsymbol{x}; \boldsymbol{\delta}) := r(\boldsymbol{x}; \boldsymbol{\delta})p(\boldsymbol{x})$:

$$\min_{\boldsymbol{\delta}} \mathrm{KL}[q|q_{\boldsymbol{\delta}}] \text{ s.t. } \int r(\boldsymbol{x}; \boldsymbol{\delta})p(\boldsymbol{x})\mathrm{d}\boldsymbol{x} = 1. \tag{1}$$

$\mathrm{KL}[q|q_{\boldsymbol{\delta}}]$ comprises three terms in which only one term is dependent on the parameter $\boldsymbol{\delta}$:

$$\mathrm{KL}[q|q_{\boldsymbol{\delta}}] = \mathbb{E}_q[\log q(\boldsymbol{x})] - \mathbb{E}_q[\log r(\boldsymbol{x}; \boldsymbol{\delta})] - \mathbb{E}_q[\log p(\boldsymbol{x})] \approx -\frac{1}{n_q} \sum_{i=1}^{n_q} \log r(\boldsymbol{x}_q^{(i)}; \boldsymbol{\delta}) + C, \tag{2}$$

The last step uses $X_q$ to approximate the expectation over $q(\boldsymbol{x})$. $C$ is a constant irrelevant to $\boldsymbol{\delta}$. We can also approximate the equality constraint in (1) using $X_p$:

$$\int r(\boldsymbol{x}; \boldsymbol{\delta})p(\boldsymbol{x})\mathrm{d}\boldsymbol{x} \approx \frac{1}{n_p} \sum_{j=1}^{n_p} r(\boldsymbol{x}_p^{(j)}; \boldsymbol{\delta}). \tag{3}$$

Combining (2) and (3), we get a sample version of (1):

$$\hat{\boldsymbol{\delta}} := \underset{\boldsymbol{\delta}}{\operatorname{argmin}} -\frac{1}{n_q} \sum_{i=1}^{n_q} \log r(\boldsymbol{x}_q^{(i)}; \boldsymbol{\delta}) + C \quad \text{s.t.} \frac{1}{n_p} \sum_{j=1}^{n_p} r(\boldsymbol{x}_p^{(j)}; \boldsymbol{\delta}) = 1. \tag{4}$$

The above optimization is called Kullback Leibler Importance Estimation Procedure (KLIEP) [30]. Unfortunately, it *cannot* be directly used to estimate our ratio $\frac{q(\boldsymbol{x})}{p(\boldsymbol{x};\boldsymbol{\theta})}$ since we only have samples from $q(\boldsymbol{x})$ but not from $p(\boldsymbol{x}; \boldsymbol{\theta})$. Consequently the equality constraint $\int r(\boldsymbol{x}; \boldsymbol{\delta})p(\boldsymbol{x}; \boldsymbol{\theta})\mathrm{d}\boldsymbol{x} = 1$ can no longer be approximated using samples.

A natural remedy to this problem is to draw samples from $p(\boldsymbol{x}; \boldsymbol{\theta})$ using sampling techniques, such as MCMC which, in general, can be costly when $p(\boldsymbol{x}; \boldsymbol{\theta})$ is complex. Correlation among drawn samples from an MCMC scheme further complicates estimation of the ratio. More importantly, regardless of the feasibility of sampling from $p(\boldsymbol{x}; \boldsymbol{\theta})$, the availability of an explicit (possibly unnormalized) density $p(\boldsymbol{x}; \boldsymbol{\theta})$ is much more valuable than just samples, especially in a high dimensional space where samples rarely capture the fine-grained structural information present in the density model $p(\boldsymbol{x}; \boldsymbol{\theta})$.

In this work, we propose a new procedure – Stein Density Ratio Estimation – which can directly use the (unnormalized) density $p$, as it is, without sampling from it. Moreover, the new procedure (described in Section 3.1) yields a density ratio model $r_{\boldsymbol{\theta}}(\boldsymbol{x}; \boldsymbol{\delta})$ for the ratio function $\frac{q(\boldsymbol{x})}{p(\boldsymbol{x};\boldsymbol{\theta})}$ that automatically satisfies the aforementioned equality constraint for all $\boldsymbol{\theta}$.

## 3 Stein Density Ratio Estimation

Let us consider a linear-in-parameter density ratio model $r(\boldsymbol{x}; \boldsymbol{\delta}) := \boldsymbol{\delta}^\top \boldsymbol{f}(\boldsymbol{x})$, where $\boldsymbol{f}(\boldsymbol{x})$ is a "feature function" that transforms a data point $\boldsymbol{x}$ into a more powerful representation. To better model $\frac{q(\boldsymbol{x})}{p(\boldsymbol{x};\boldsymbol{\theta})}$, we define a family of feature functions called Stein features.

### 3.1 Stein Features

Suppose we have a feature function $\boldsymbol{f}(\boldsymbol{x}) : \mathbb{R}^d \to \mathbb{R}^b$ and a density model $p(\boldsymbol{x}; \boldsymbol{\theta}) : \mathbb{R}^d \to \mathbb{R}$. A Stein feature $T_{\boldsymbol{\theta}}\boldsymbol{f}(\boldsymbol{x}) \in \mathbb{R}^b$ with respect to $p(\boldsymbol{x}; \boldsymbol{\theta})$ is $T_{\boldsymbol{\theta}}\boldsymbol{f}(\boldsymbol{x}) := [T_{\boldsymbol{\theta}}f_1(\boldsymbol{x}), \ldots, T_{\boldsymbol{\theta}}f_i(\boldsymbol{x}), \ldots, T_{\boldsymbol{\theta}}f_b(\boldsymbol{x})]^\top$, where $T_{\boldsymbol{\theta}}$ is a *Stein operator* [29, 8, 4, 23] and $T_{\boldsymbol{\theta}}f_i(\boldsymbol{x}) \in \mathbb{R}$ is defined as

$$T_{\boldsymbol{\theta}}f_i(\boldsymbol{x}) := \langle \nabla_{\boldsymbol{x}} \log p(\boldsymbol{x}; \boldsymbol{\theta}), \nabla_{\boldsymbol{x}} f_i(\boldsymbol{x}) \rangle + \mathrm{trace}(\nabla_{\boldsymbol{x}}^2 f_i(\boldsymbol{x})),$$

where $f_i$ is the $i$-th output of function $\boldsymbol{f}$, $\nabla_{\boldsymbol{x}} f_i(\boldsymbol{x})$ is the gradient of $f_i(\boldsymbol{x})$ and $\nabla_{\boldsymbol{x}}^2 f_i(\boldsymbol{x})$ is the Hessian of $f_i(\boldsymbol{x})$. Note that computing $T_{\boldsymbol{\theta}} \boldsymbol{f}(\boldsymbol{x})$ does *not* require evaluating the normalization term $Z(\boldsymbol{\theta})$ as

$$\nabla_{\boldsymbol{x}} \log p(\boldsymbol{x}; \boldsymbol{\theta}) = \nabla_{\boldsymbol{x}} \log \bar{p}(\boldsymbol{x}; \boldsymbol{\theta}) - \nabla_{\boldsymbol{x}} \log Z(\boldsymbol{\theta}), \text{ where } \nabla_{\boldsymbol{x}} \log Z(\boldsymbol{\theta}) = 0.$$

**Example 1.** *Let $p(\boldsymbol{x}; \boldsymbol{\theta})$ be in exponential family with sufficient statistic $\boldsymbol{\psi}(\boldsymbol{x})$, then*

$$T_{\boldsymbol{\theta}} f_i(\boldsymbol{x}) = \boldsymbol{\theta}^\top \boldsymbol{J}_{\boldsymbol{x}} \boldsymbol{\psi}(\boldsymbol{x}) \nabla_{\boldsymbol{x}} f_i(\boldsymbol{x}) + \text{trace}[\nabla_{\boldsymbol{x}}^2 f_i(\boldsymbol{x})],$$

*where $\boldsymbol{J}_{\boldsymbol{x}} \boldsymbol{\psi}(\boldsymbol{x}) \in \mathbb{R}^{\dim(\boldsymbol{\theta}) \times d}$ is the Jacobian of $\boldsymbol{\psi}(\boldsymbol{x})$ and $\dim(\boldsymbol{\theta})$ is the dimension of $\boldsymbol{\theta}$ .*

One more example can be found at Appendix, Section A.1. A slightly different Stein operator was introduced in [4, 23] where $T'_{\boldsymbol{\theta}} \boldsymbol{f}(\boldsymbol{x}) \in \mathbb{R}$ for $\boldsymbol{f}(\boldsymbol{x}) \in \mathbb{R}^d$ is defined as $T'_{\boldsymbol{\theta}} \boldsymbol{f}(\boldsymbol{x}) := \sum_{i=1}^d [\partial_{x_i} \log p(\boldsymbol{x}; \boldsymbol{\theta})] f_i(\boldsymbol{x}) + \partial_{x_i} f_i(\boldsymbol{x})$, where $\partial_{x_i} f(\boldsymbol{x})$ is the partial derivative of $f(\boldsymbol{x})$ with respect to $x_i$. We can see the relationship between $T$ and $T'$: $T_{\boldsymbol{\theta}} f_i(\boldsymbol{x}) = T'_{\boldsymbol{\theta}} \nabla_{\boldsymbol{x}} f_i(\boldsymbol{x})$. Next we show an important property of Stein features.

**Proposition 1** (Stein's Identity)**.** *Suppose $p(\boldsymbol{x}; \boldsymbol{\theta}) > 0$,*

$$\forall_{i,j} \lim_{|x_j| \to \infty} p(x_1, \cdots, x_j, \cdots, x_d; \boldsymbol{\theta}) \partial_{x_j} f_i(x_1, \cdots, x_j, \cdots, x_d) = 0,$$

*$p(\boldsymbol{x}; \boldsymbol{\theta})$ is continuously differentiable and $f_i$ is twice continuously differentiable for all $\boldsymbol{\theta}$ and $i$. Then $\mathbb{E}_{p_{\boldsymbol{\theta}}}[T_{\boldsymbol{\theta}} \boldsymbol{f}(\boldsymbol{x})] = \boldsymbol{0}$ for all $\boldsymbol{\theta}$.*

We give a proof in Appendix Section B.1. Similar identities were given in previous literatures such as Lemma 2.2 in [19] or Lemma 5.1 in [4]. Utilizing this property, we can construct a density ratio model which bypasses the "intractable equality constraint" issue when estimating $\frac{q(\boldsymbol{x})}{p(\boldsymbol{x}; \boldsymbol{\theta})}$ as shown in the next section.

## 3.2 Stein Density Ratio Modeling and Estimation (SDRE)

Define a linear-in-parameter density ratio model: $r_{\boldsymbol{\theta}}(\boldsymbol{x}; \boldsymbol{\delta}) := \boldsymbol{\delta}^\top T_{\boldsymbol{\theta}} \boldsymbol{f}(\boldsymbol{x}) + 1$ by using a Stein feature function. We can see that $\mathbb{E}_{p_{\boldsymbol{\theta}}}[r_{\boldsymbol{\theta}}(\boldsymbol{x}; \boldsymbol{\delta})] = \mathbb{E}_{p_{\boldsymbol{\theta}}}[\boldsymbol{\delta}^\top T_{\boldsymbol{\theta}} \boldsymbol{f}(\boldsymbol{x}) + 1] = \boldsymbol{\delta}^\top \mathbb{E}_{p_{\boldsymbol{\theta}}}[T_{\boldsymbol{\theta}} \boldsymbol{f}(\boldsymbol{x})] + 1 = 1$ where the last equality is ensured by Proposition 1 for all $\boldsymbol{\delta}$ and $\boldsymbol{\theta}$ if the specified regularity conditions are met. This equality means the constraint in (1) is automatically satisfied with this density ratio model. Now we can solve (4) without its equality constraint.

$$\hat{\boldsymbol{\delta}} := \operatorname*{argmin}_{\boldsymbol{\delta} \in \mathbb{R}^b} -\frac{1}{n_q} \sum_{i=1}^{n_q} \log r_{\boldsymbol{\theta}}(\boldsymbol{x}_q^{(i)}; \boldsymbol{\delta}) + C = \operatorname*{argmax}_{\boldsymbol{\delta} \in \mathbb{R}^b} \frac{1}{n_q} \sum_{i=1}^{n_q} \log \left[ \boldsymbol{\delta}^\top T_{\boldsymbol{\theta}} \boldsymbol{f}(\boldsymbol{x}_q^{(i)}) + 1 \right]. \quad (5)$$

It can be seen that (5) is an unconstrained *concave* maximization problem. Note for all $\boldsymbol{x}_q \in X_q$, $r_{\boldsymbol{\theta}}(\boldsymbol{x}_q; \hat{\boldsymbol{\delta}})$ *must be strictly positive* thanks to the *log-barrier* (see e.g., Section 17.2 in [22]) in our objective function. However, it is *not* possible to guarantee that for all $\boldsymbol{x} \in \mathbb{R}^d$, $r_{\boldsymbol{\theta}}(\boldsymbol{x}; \hat{\boldsymbol{\delta}})$ is positive. This is not a problem in this paper, as the density ratio function is only used for approximating the KL divergence, and we will not evaluate $r_{\boldsymbol{\theta}}(\boldsymbol{x}; \hat{\boldsymbol{\delta}})$ at a data point $\boldsymbol{x}$ that is outside of $X_q$. Note, the unnormalized density model $\bar{p}(\boldsymbol{x}; \boldsymbol{\theta})$, by definition, should be non-negative everywhere for all $\boldsymbol{\theta}$.

We refer to the objective (5) as Stein Density Ratio Estimation (SDRE). One may notice that $\frac{1}{n_q} \sum_{i=1}^{n_q} \log r_{\boldsymbol{\theta}}(\boldsymbol{x}_q^{(i)}; \hat{\boldsymbol{\delta}})$ evaluated at $\hat{\boldsymbol{\delta}}$ is exactly the sample average of the estimated ratio over $X_q$ which allows us to approximate the KL divergence from $p(\boldsymbol{x}; \boldsymbol{\theta})$ to $q(\boldsymbol{x})$.

## 4 Intractable Model Inference via Discriminative Likelihood Estimation

Let $\ell(\hat{\boldsymbol{\delta}}, \boldsymbol{\theta}) := \frac{1}{n_q} \sum_{i=1}^{n_q} \log r_{\boldsymbol{\theta}}(\boldsymbol{x}_q^{(i)}; \hat{\boldsymbol{\delta}})$. We will use $\ell(\hat{\boldsymbol{\delta}}, \boldsymbol{\theta})$ as a replacement of $\text{KL}[q(\boldsymbol{x})|p(\boldsymbol{x}; \boldsymbol{\theta})]$. The rationale of minimizing KL divergence from $p(\boldsymbol{x}; \boldsymbol{\theta})$ to $q(\boldsymbol{x})$ leads to:

$$\min_{\boldsymbol{\theta}} \ell(\hat{\boldsymbol{\delta}}, \boldsymbol{\theta}) \text{ or equivalently } \min_{\boldsymbol{\theta}} \max_{\boldsymbol{\delta}} \ell(\boldsymbol{\delta}, \boldsymbol{\theta}). \quad (6)$$

The equivalence is due to the fact that $\ell$ evaluated at the optimal ratio parameter $\hat{\boldsymbol{\delta}}$ is also the maximum of the DRE objective function when being optimized w.r.t. $\boldsymbol{\delta}$. The outer problem minimizes $\ell$ with respect to the density parameter $\boldsymbol{\theta}$. We call this estimator **Discriminative Likelihood Estimation** (DLE) as the parameter of the density model $p(\boldsymbol{x}; \boldsymbol{\theta})$ is learned via minimizing a *discriminator*[1], which is the likelihood ratio function $\ell(\hat{\boldsymbol{\delta}}, \boldsymbol{\theta})$ measuring the differences between $q(\boldsymbol{x})$ and $p(\boldsymbol{x}; \boldsymbol{\theta})$.

## 4.1 Consistency with Correct Model

For brevity, we state all theorems assuming all regularity conditions in Proposition 1 are met.

**Notations:** $\boldsymbol{H}$ is $\nabla^2_{(\boldsymbol{\delta}, \boldsymbol{\theta})} \ell(\boldsymbol{\delta}, \boldsymbol{\theta})$, the full Hessian of $\ell(\boldsymbol{\delta}, \boldsymbol{\theta})$. $\boldsymbol{H}_{\boldsymbol{\delta}, \boldsymbol{\theta}}$ is $\nabla_{\boldsymbol{\delta}} \nabla_{\boldsymbol{\theta}} \ell(\boldsymbol{\delta}, \boldsymbol{\theta})$, submatrix of the Hessian matrix whose rows and columns indexed by $\boldsymbol{\delta}, \boldsymbol{\theta}$ respectively. $\boldsymbol{s} \in \mathbb{R}^{\dim(\boldsymbol{\theta})}$ is $\nabla_{\boldsymbol{\theta}} \log p(\boldsymbol{x}, \boldsymbol{\theta})$ evaluated at $\boldsymbol{\theta}^*$, score function of $p(\boldsymbol{x}; \boldsymbol{\theta})$. $\lambda(\cdot)$ is the eigenvalue operator. $\lambda_{\min}(\cdot)$ or $\lambda_{\max}(\cdot)$ is the minimum or maximum eigenvalue and $\|\cdot\|$ is the operator norm.

We study the consistency of the following estimator under a correct model setting.

$$(\hat{\boldsymbol{\delta}}, \hat{\boldsymbol{\theta}}) := \arg \min_{\boldsymbol{\theta} \in \Theta} \max_{\boldsymbol{\delta} \in \Delta_{n_q}} \ell(\boldsymbol{\delta}, \boldsymbol{\theta}), \tag{7}$$

where $\Theta$ and $\Delta_{n_q}$ are *compact* parameter spaces for $\boldsymbol{\theta}$ and $\boldsymbol{\delta}$ respectively. The compactness condition is among a set of conditions commonly used in classic consistency proofs (see e.g., Wald's Consistency Proof, 5.2.1,[32]). It is possible to derive weaker conditions given specific choices of $\boldsymbol{f}$ or $p(\boldsymbol{x}; \boldsymbol{\theta})$. However, in the current manuscript, we only focus on more generic settings and conditions that would give rise to estimation consistency and useful asymptotic theories. We assume they are properly chosen so that $(\hat{\boldsymbol{\theta}}, \hat{\boldsymbol{\delta}})$ is the saddle point of (7).

First, we assume our density model $p(\boldsymbol{x}; \boldsymbol{\theta})$ is correctly specified:

**Assumption 1.** *There exists a unique pair of parameter* $(\boldsymbol{\theta}^*, \boldsymbol{\delta}^*), \boldsymbol{\theta}^* \in \Theta, \boldsymbol{\delta}^* \in \Delta_{n_q}$, *such that* $p(\boldsymbol{x}; \boldsymbol{\theta}^*) \equiv q(\boldsymbol{x})$ *and* $r_{\boldsymbol{\theta}^*}(\boldsymbol{x}; \boldsymbol{\delta}^*) = 1$.

Given how $r_{\boldsymbol{\theta}}(\boldsymbol{x}; \boldsymbol{\delta})$ is constructed in Section 3.2, the above assumption implies $\boldsymbol{\delta}^*$ must be $\boldsymbol{0}$.

**Assumption 2.** *There exist constants* $\Lambda_{\min} > 0, \Lambda'_{\min} > 0$ *and* $\Lambda_{\max} > 0$ *so that* $\forall \boldsymbol{\theta} \in \Theta, \boldsymbol{\delta} \in \Delta_{n_q}$

$$\lambda_{\min} \left\{ -\boldsymbol{H}_{\boldsymbol{\delta}, \boldsymbol{\delta}} \right\} \geq \Lambda'_{\min}, \Lambda_{\max} \geq \left\| \boldsymbol{H}_{\boldsymbol{\theta}, \boldsymbol{\delta}} \boldsymbol{H}_{\boldsymbol{\delta}, \boldsymbol{\delta}}^{-1} \right\|, \lambda_{\min} \left\{ -\boldsymbol{H}_{\boldsymbol{\theta}, \boldsymbol{\delta}} \boldsymbol{H}_{\boldsymbol{\delta}, \boldsymbol{\delta}}^{-1} \boldsymbol{H}_{\boldsymbol{\delta}, \boldsymbol{\theta}} \right\} \geq \Lambda_{\min} > 2 \left\| \boldsymbol{H}_{\boldsymbol{\theta}, \boldsymbol{\theta}} \right\|.$$

The lower-boundedness of $\lambda_{\min} \left\{ -\boldsymbol{H}_{\boldsymbol{\delta}, \boldsymbol{\delta}} \right\}$ implies the *strict* concavity of $\ell(\boldsymbol{\delta}, \boldsymbol{\theta})$ with respect to $\boldsymbol{\delta}$ ($\ell(\boldsymbol{\delta}, \boldsymbol{\theta})$ is already concave by construction, see (5)): For all $\boldsymbol{\theta} \in \Theta$, there exists a unique $\hat{\boldsymbol{\delta}}(\boldsymbol{\theta})$ that maximizes the likelihood ratio, which means the likelihood ratio function should always have sufficient discriminative power to precisely pinpoint the differences between our data and the current model $\boldsymbol{\theta}$. It also ensures that $\boldsymbol{\delta}$ can "teach" the model parameter $\boldsymbol{\theta}$ well by assuming the "interaction" between $\boldsymbol{\delta}$ and $\boldsymbol{\theta}$ in our estimator, $\boldsymbol{H}_{\boldsymbol{\theta}, \boldsymbol{\delta}}$, is well-behaved.

Now we analyze Assumption 2 on a special case:

**Proposition 2.** *Let* $\Delta_{n_q} := \left\{ \boldsymbol{\delta} \, \middle| \, \frac{1}{C_{\mathrm{ratio}}} \leq r_{\boldsymbol{\theta}}(\boldsymbol{x}; \boldsymbol{\delta}) \leq C_{\mathrm{ratio}}, \|\boldsymbol{\delta}\|_2 \leq T/\sigma(n_q), \forall \boldsymbol{\theta} \in \Theta, \forall \boldsymbol{x} \in X_q, \right\}$
*where* $T > 0, C_{\mathrm{ratio}} > 1$ *are constants and* $\sigma(\cdot)$ *is a monotone-increasing function.* $p(\boldsymbol{x}; \boldsymbol{\theta})$ *is in exponential family with sufficient statistic* $\boldsymbol{\psi}(\boldsymbol{x})$ *and Stein feature is chosen as* $T_{\boldsymbol{\theta}} \boldsymbol{\psi}(\boldsymbol{x})$. *Suppose there exist constants* $C_2, C_3, C_4, C_5, \Lambda''_{\max}, \Lambda''_{\min} > 0, C_2 \geq \frac{1}{n_q} \sum_{i=1}^{n_q} \|\boldsymbol{J}_{\boldsymbol{x}} \boldsymbol{\psi}(\boldsymbol{x}_q^{(i)})\|^4$,

$$\lambda_{\min} \left\{ \frac{1}{n_q} \sum_{i=1}^{n_q} \boldsymbol{J}_{\boldsymbol{x}} \boldsymbol{\psi}(\boldsymbol{x}_q^{(i)}) \boldsymbol{J}_{\boldsymbol{x}} \boldsymbol{\psi}(\boldsymbol{x}_q^{(i)})^\top \right\} \geq C_3, \frac{1}{n_q} \sum_{i=1}^{n_q} \|\boldsymbol{J}_{\boldsymbol{x}} \boldsymbol{\psi}(\boldsymbol{x}_q^{(i)}) \boldsymbol{J}_{\boldsymbol{x}} \boldsymbol{\psi}(\boldsymbol{x}_q^{(i)})^\top\| \leq C_4,$$

$$\frac{1}{n_q} \sum_{i=1}^{n_q} \|\boldsymbol{J}_{\boldsymbol{x}} \boldsymbol{\psi}(\boldsymbol{x}_q^{(i)}) \boldsymbol{J}_{\boldsymbol{x}} \boldsymbol{\psi}(\boldsymbol{x}_q^{(i)})^\top\| \cdot \|T_{\boldsymbol{\theta}} \boldsymbol{\psi}(\boldsymbol{x}_q^{(i)})\| \leq C_5$$

*and* $\Lambda''_{\max} \geq \lambda \left( \frac{1}{n_q} \sum_{i=1}^{n_q} T_{\boldsymbol{\theta}} \boldsymbol{\psi}(\boldsymbol{x}_q^{(i)}) T_{\boldsymbol{\theta}} \boldsymbol{\psi}(\boldsymbol{x}_q^{(i)})^\top \right) \geq \Lambda''_{\min}, \forall \boldsymbol{\theta} \in \Theta$ *with high probability. There exists a constant* $N > 1$, *when* $n_q \geq N$, *Assumption 2 holds with high probability.*

The proof can be found in Appendix, Section B.3. Note in practice the domain constraint of $\Delta_{n_q}$ in this proposition can be easily enforced via convex constraints or penalty terms. Analysis on a few other examples can be found in Appendix, Section A.2.

Proposition 2 gives us some hints on how the feature function $\boldsymbol{f}$ of Stein feature can be chosen. In the case of exponential family, the choice $\boldsymbol{f} = \boldsymbol{\phi}$ guarantees Assumption 2 to hold with high probability when $n_q$ increases.

**Assumption 3** (Concentration of Stein features). *The difference between the sample average of the Stein feature $T_{\boldsymbol{\theta}^*}\boldsymbol{f}(\boldsymbol{x})$ and its expectation over $q$ converges to zero in $\ell_2$ norm in probability.*
$$\left\| \frac{1}{n_q} \sum_{i=1}^{n_q} T_{\boldsymbol{\theta}^*}\boldsymbol{f}(\boldsymbol{x}_q^{(i)}) - \mathbb{E}_q\left[T_{\boldsymbol{\theta}^*}\boldsymbol{f}(\boldsymbol{x})\right] \right\|_2 \xrightarrow{\mathbb{P}} 0.$$

Note, if Assumption 1 holds at the same time, Proposition 1 indicates $\mathbb{E}_q\left[T_{\boldsymbol{\theta}^*}\boldsymbol{f}(\boldsymbol{x})\right] \equiv \boldsymbol{0}$. This assumption holds due to the (strong) law of large numbers given that the $\mathbb{E}_q\left[T_{\boldsymbol{\theta}^*}\boldsymbol{f}(\boldsymbol{x})\right]$ exists.

**Theorem 1** (Consistency). *Suppose Assumption 1, 2 and 3 holds, $(\hat{\boldsymbol{\delta}}, \hat{\boldsymbol{\theta}}) \xrightarrow{\mathbb{P}} (\boldsymbol{0}, \boldsymbol{\theta}^*)$.*

See Section B.4 in Appendix for the proof. This theorem states that as $n_q$ increases, all saddle points of (7), converge to the vicinity of true parameters. All the following theorems rely on the result of Theorem 1.

## 4.2 Asymptotic Variance of $\hat{\theta}$ and Fisher Efficiency of DLE

In this section we state one of our main contributions: DLE can attain the efficiency bound, i.e., asymptotic Cramér-Rao bound when $\boldsymbol{f}(\boldsymbol{x})$ is appropriately chosen. First, we show our estimator $\hat{\boldsymbol{\theta}}$ has a simple asymptotic distribution which allows us to perform model inference. To state the theorem, we need an extra assumption on the Hessian $\boldsymbol{H}$:

**Assumption 4** (Uniform Convergence on $\boldsymbol{H}$). $\sup_{\boldsymbol{\delta} \in \Delta_{n_q}, \boldsymbol{\theta} \in \Theta} |H_{i,j} - \mathbb{E}_q[H_{i,j}]| \xrightarrow{\mathbb{P}} 0, \forall_{i,j}.$

This assumption states the second order derivatives (which is an average over samples from $X_q$) converges **uniformly** to its population mean, as $n_q \to \infty$. It helps us control the residual in the second order Taylor expansion in our proof. This assumption may be weakened given specific choices of $\boldsymbol{f}$ and $p(\boldsymbol{x}; \boldsymbol{\theta})$ but we focus on establishing the asymptotic results in generic settings, so this condition is only listed as an assumption.

**Theorem 2** (Asymptotic Normality of $\hat{\boldsymbol{\theta}}$). *Suppose Assumption 1, 2, 3 and 4 holds,*

$$\sqrt{n_q}\left(\boldsymbol{\theta}^* - \hat{\boldsymbol{\theta}}\right) \rightsquigarrow \mathcal{N}\left[0, \boldsymbol{V}\right], \boldsymbol{V} = \left(-\mathbb{E}_q\left[\boldsymbol{H}_{\boldsymbol{\theta}, \boldsymbol{\delta}}^*\right] \mathbb{E}_q\left[\boldsymbol{H}_{\boldsymbol{\delta}, \boldsymbol{\delta}}^*\right]^{-1} \mathbb{E}_q\left[\boldsymbol{H}_{\boldsymbol{\delta}, \boldsymbol{\theta}}^*\right]\right)^{-1}, \quad (8)$$

*where $\boldsymbol{H}^*$ is $\boldsymbol{H}$ evaluated at $(\boldsymbol{\delta}^*, \boldsymbol{\theta}^*)$.*

See Section B.5 in Appendix for the proof. In practice, we do not know $\mathbb{E}_q\left[\boldsymbol{H}^*\right]$, so we may use $\hat{\boldsymbol{H}}$, the Hessian of $\ell(\boldsymbol{\delta}, \boldsymbol{\theta})$ evaluated at $(\hat{\boldsymbol{\delta}}, \hat{\boldsymbol{\theta}})$ as an approximation to $\mathbb{E}_q\left[\boldsymbol{H}^*\right]$.

Although MLE is also asymptotically normal, important quantities such as Fisher Information Matrix may not be efficiently computed on intractable models. In comparison, Theorem 2 enables us to compute parameter confidence interval for DLE even on intractable $p_{\boldsymbol{\theta}}$.

Now we consider the asymptotic efficiency of the DLE with respect to specific choices of Stein features. Let $\boldsymbol{V}_{\boldsymbol{f}}$ be the asymptotic variance (8) using a Stein feature with a specific choice of $\boldsymbol{f}$.

**Lemma 3.** *Suppose that Assumptions 1, 2, 3 and 4 hold and $\mathbb{E}_q[T_{\boldsymbol{\theta}^*}\boldsymbol{f}(\boldsymbol{x})T_{\boldsymbol{\theta}^*}\boldsymbol{f}(\boldsymbol{x})^\top]$ is invertible. Moreover, suppose that the integration and the derivative of $\partial_{\theta_i} \int p(\boldsymbol{x}; \boldsymbol{\theta})T_{\boldsymbol{\theta}}\boldsymbol{f}(\boldsymbol{x})\mathrm{d}\boldsymbol{x}$ is exchangeable for all $i$. $\boldsymbol{V}_{\boldsymbol{f}} = \left(\mathbb{E}_q[\boldsymbol{s}T_{\boldsymbol{\theta}^*}\boldsymbol{f}(\boldsymbol{x})^\top]\mathbb{E}_q[T_{\boldsymbol{\theta}^*}\boldsymbol{f}(\boldsymbol{x})T_{\boldsymbol{\theta}^*}\boldsymbol{f}(\boldsymbol{x})^\top]^{-1}\mathbb{E}_q[T_{\boldsymbol{\theta}^*}\boldsymbol{f}(\boldsymbol{x})\boldsymbol{s}^\top]\right)^{-1}.$*

The proof is given in Section B.6 in the Appendix. Lemma 3 expresses asymptotic variance using score function and Stein feature and is used to prove that the variance monotonically decreases as the vector space spanned by the Stein feature vectors becomes larger.

**Corollary 4** (Monotonocity of Asymptotic Variance). *Define the inner product as $\mathbb{E}_q[fg]$ for functions $f$ and $g$. Let $T_{\boldsymbol{\theta}^*}\boldsymbol{f}(\boldsymbol{x}) = [t_1, \ldots, t_b]$ and $T_{\boldsymbol{\theta}^*}\bar{\boldsymbol{f}}(\boldsymbol{x}) = [\bar{t}_1, \ldots, \bar{t}_{\bar{b}}]$ be two Stein feature vectors. Assume that $\mathrm{span}\{t_1, \ldots, t_b\} \subset \mathrm{span}\{\bar{t}_1, \ldots, \bar{t}_{\bar{b}}\}$, where $\mathrm{span}\{\cdots\}$ denotes the linear space*

*spanned by the specified elements. Then, the inequality $V_{\bar{\boldsymbol{f}}} \preceq V_{\boldsymbol{f}}$ holds in the sense of the positive definiteness.*

*Proof.* Let us define $P_{\boldsymbol{f}}\boldsymbol{s}$ as the orthogonal projection of $\boldsymbol{s}$ onto $\mathrm{span}\{t_1, \ldots, t_b\}$. A simple calculation yields $P_{\boldsymbol{f}}\boldsymbol{s} = \mathbb{E}_q[\boldsymbol{s}T_{\boldsymbol{\theta}^*}\boldsymbol{f}(\boldsymbol{x})^{\top}]\mathbb{E}_q[T_{\boldsymbol{\theta}^*}\boldsymbol{f}(\boldsymbol{x})T_{\boldsymbol{\theta}^*}\boldsymbol{f}(\boldsymbol{x})^{\top}]^{-1}T_{\boldsymbol{\theta}^*}\boldsymbol{f}(\boldsymbol{x})$, and thus, Lemma 3 leads to $V_{\boldsymbol{f}}^{-1} = \mathbb{E}_q[P_{\boldsymbol{f}}\boldsymbol{s}(P_{\boldsymbol{f}}\boldsymbol{s})^{\top}]$. From the property of the orthogonal projection (see e.g., Theorem 2.23 in [34]), we have $\mathbb{E}_q[P_{\bar{\boldsymbol{f}}}\boldsymbol{s}(P_{\bar{\boldsymbol{f}}}\boldsymbol{s})^{\top}] \succeq \mathbb{E}_q[P_{\boldsymbol{f}}\boldsymbol{s}(P_{\boldsymbol{f}}\boldsymbol{s})^{\top}]$. Therefore, we obtain $V_{\bar{\boldsymbol{f}}} \preceq V_{\boldsymbol{f}}$. $\qquad\square$

For $Q_{\boldsymbol{f}}\boldsymbol{s} = \boldsymbol{s} - P_{\boldsymbol{f}}\boldsymbol{s}$, we have $\mathbb{E}_q[\boldsymbol{s}\boldsymbol{s}^{\top}] = \mathbb{E}_q[P_{\boldsymbol{f}}\boldsymbol{s}(P_{\boldsymbol{f}}\boldsymbol{s})^{\top}] + \mathbb{E}_q[Q_{\boldsymbol{f}}\boldsymbol{s}(Q_{\boldsymbol{f}}\boldsymbol{s})^{\top}] = V_{\boldsymbol{f}}^{-1} + \mathbb{E}_q[Q_{\boldsymbol{f}}\boldsymbol{s}(Q_{\boldsymbol{f}}\boldsymbol{s})^{\top}]$. Thus, we see that the asymptotic variance converges to the inverse of the Fisher information, $\mathbb{E}_q[\boldsymbol{s}\boldsymbol{s}^{\top}]^{-1}$, as $P_{\boldsymbol{f}}\boldsymbol{s}$ gets close to $\boldsymbol{s}$. In particular, when the linear space $\mathrm{span}\{t_1, \ldots, t_b\}$ includes $\boldsymbol{s}$, $Q_{\boldsymbol{f}}\boldsymbol{s}$ vanishes and consequently the DLE with $\boldsymbol{f}(\boldsymbol{x})$ is asymptotically efficient.

**Example 2.** *Let $p(\boldsymbol{x}; \boldsymbol{\theta})$ be the model of the $d$-dimensional multivariate Gaussian distribution $\mathcal{N}(\boldsymbol{\theta}, \mathrm{Iden} \cdot \sigma^2)$, where $\mathrm{Iden}$ is the identity matrix. Here the variance $\sigma^2$ is assumed to be known. The score function is $s_j(\boldsymbol{x}; \boldsymbol{\theta}) = -(x_j - \theta_j)/\sigma^2$, and the Stein feature vector defined from $\boldsymbol{f}(\boldsymbol{x}) = \boldsymbol{x}$ is $(T_{\boldsymbol{\theta}}\boldsymbol{x})_j = -(x_j - \theta_j)/\sigma^2$ for $j = 1, \ldots, d$. Clearly, the score function is included in $\mathrm{span}\{t_1, \ldots, t_d\}$. Hence, the DLE with $\boldsymbol{f}$ achieves the efficiency bound of the parameter estimation.*

One more example can be found in Appendix, Section A.3. In fact, Corollary 3 suggests that as long as we can represent the score function $\boldsymbol{s}$ using Stein feature $T_{\boldsymbol{\theta}}\boldsymbol{f}$ up to a linear transformation, DLE can achieve efficiency bound. However, since $\boldsymbol{f}$ is coupled with $\nabla_{\boldsymbol{x}}\log p(\boldsymbol{x}; \boldsymbol{\theta})$ in $T_{\boldsymbol{\theta}}\boldsymbol{f}$, it is not always easy to reverse engineer an $\boldsymbol{f}$ from $\boldsymbol{s}$. Nonetheless, our numerical experiments show that using simple functions such polynomials as $\boldsymbol{f}$ yields good performance.

### 4.3 Model Selection of DLE

As our objective (6) tries to minimize the discrepancy between our model $p(\boldsymbol{x}; \boldsymbol{\theta})$ and the data distribution, it is tempting to compare models using the objective function evaluated at $(\hat{\boldsymbol{\delta}}, \hat{\boldsymbol{\theta}})$, i.e., $\ell(\hat{\boldsymbol{\delta}}, \hat{\boldsymbol{\theta}})$. However, the more sophisticated $p(\boldsymbol{x}; \boldsymbol{\theta})$ becomes, the more likely it picks up spurious patterns of our dataset. Similarly, the more powerful the Stein features are, the more likely the discriminator is overly critical to the density model on this dataset. Thus a better model selection criterion would be comparing $\mathbb{E}_q\left[\ell(\hat{\boldsymbol{\delta}}, \hat{\boldsymbol{\theta}})\right]$ which eliminates the potential of overfitting a specific dataset. Unfortunately, this expectation cannot be computed without the knowledge on $q(\boldsymbol{x})$. We propose to approximate this quantity using a *penalized likelihood*:

**Theorem 5.** *Suppose Assumption 1, 2, 3 and 4 holds. $\mathbb{E}_q\left[\boldsymbol{H}_{\boldsymbol{\delta}, \boldsymbol{\delta}}^*\right]$ and $\mathbb{E}_q\left[\boldsymbol{H}_{\boldsymbol{\delta}, \boldsymbol{\theta}}^*\right]$ are full-rank and $\dim(\boldsymbol{\theta}) \leq b$, then $n_q\mathbb{E}_q\left[\ell(\hat{\boldsymbol{\delta}}, \hat{\boldsymbol{\theta}})\right] = \min_{\boldsymbol{\theta}} \max_{\boldsymbol{\delta}} n_q\ell(\boldsymbol{\delta}, \boldsymbol{\theta}) - b + \dim(\boldsymbol{\theta}) + o_p(1)$.*

See Section B.7 in Appendix for the proof. This theorem is closely related to a classic result called Akaike Information Criterion (AIC) [1]. Both AIC and Theorem 5 similarly penalize the degree of freedom of the density model $\dim(\boldsymbol{\theta})$, while our theorem also penalizes the number of ratio parameter $\dim(\boldsymbol{\delta}) = b$ due to the fact that our ratio function is also fitted using samples.

One can also show $\ell(\hat{\boldsymbol{\delta}}, \hat{\boldsymbol{\theta}})$ follows a $\chi^2$ distribution. See Section B.8 in Appendix for details.

Theorem 5 provides an information-criterion based model selection method. Suppose $M$ is a set of different Stein features and $M'$ is a set of candidate density models. We can jointly select density model and Stein feature: $(\hat{m}, \hat{m}') := \arg\min_{m' \in M'} \max_{m \in M} \mathbb{E}_q[\ell(\hat{\boldsymbol{\theta}}(m'), \hat{\boldsymbol{\delta}}(m))]$, where $(\hat{\boldsymbol{\theta}}(m'), \hat{\boldsymbol{\delta}}(m))$ are estimated parameters under the model choice $(m', m)$. Replacing $\mathbb{E}_q[\ell(\hat{\boldsymbol{\theta}}(m'), \hat{\boldsymbol{\delta}}(m))]$ with the penalized likelihood derived in Theorem 5, we can get a practical model selection method.

## 5 Lagrangian Dual of SDRE and DLE by Minimization

Some techniques can be used to directly optimize the min-max problem in (6), such as performing gradient descend/ascend on $\boldsymbol{\theta}$ and $\boldsymbol{\delta}$ alternately. However, looking for the saddle points of a min-max optimization is hard. In this section, we derive a partial Lagrangian dual for (6) so we can convert this

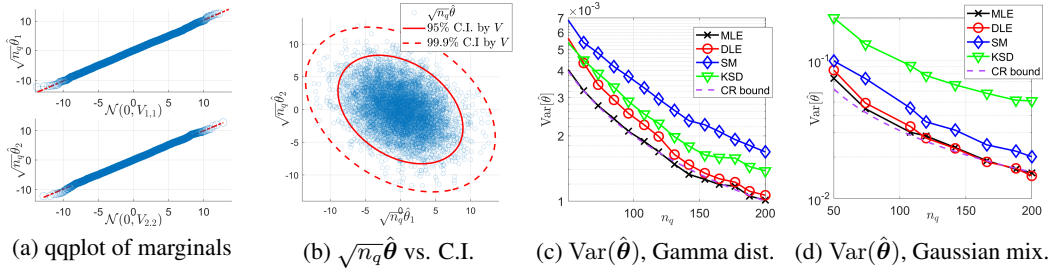

|  |  |  |  |
|---|---|---|---|
| (a) qqplot of marginals | (b) $\sqrt{n_q}\hat{\boldsymbol{\theta}}$ vs. C.I. | (c) $\mathrm{Var}(\hat{\boldsymbol{\theta}})$, Gamma dist. | (d) $\mathrm{Var}(\hat{\boldsymbol{\theta}})$, Gaussian mix. |

Figure 1: Theoretical Prediction values vs. Empirical results

min-max problem into a min-min problem whose local optima can be efficiently found by existing optimization techniques.

**Proposition 3.** *SDRE problem in* (5) *has a Lagrangian dual:*

$$\hat{\boldsymbol{\mu}} = \operatorname*{argmin}_{\boldsymbol{\mu}} \sum_{i=1}^{n_q}[-(\log -\mu_i) - 1] - \sum_{i=1}^{n_q} \mu_i \ \ \text{s.t.} : \sum_{i=1}^{n_q} \mu_i T_{\boldsymbol{\theta}} \boldsymbol{f}(\boldsymbol{x}_q^{(i)}) = \boldsymbol{0}. \tag{9}$$

*Moreover, the duality gap between* (9) *and* (5) *is 0 and* $r_{\boldsymbol{\theta}}(\boldsymbol{x}_q^{(i)}; \hat{\boldsymbol{\delta}}) = -1/\hat{\mu}_i$.

See Section B.9 in the Appendix for its proof. Instead of solving the min-max problem (6), we solve the following constrained minimization problem:

$$\min_{\boldsymbol{\theta}} \min_{\boldsymbol{\mu}} \sum_{i=1}^{n_q}[-(\log -\mu_i) - 1] - \sum_{i=1}^{n_q} \mu_i, \ \ \text{s.t.} : \sum_{i=1}^{n_q} \mu_i T_{\boldsymbol{\theta}} \boldsymbol{f}(\boldsymbol{x}_q^{(i)}) = \boldsymbol{0}, \tag{10}$$

where we replace the inner max problem in (6) with its Lagrangian (9). All experiments in this paper are performed using the Lagrangian dual objective (10). See `https://github.com/lamfeeling/Stein-Density-Ratio-Estimation` for code demos on SDRE and model inference.

## 6 Related Works

**Score Matching (SM) [13, 14]** is a inference method for unnormalized statistical models. It estimates model parameters by minimizing the Fisher divergence [20, 26] between the true log density and the model log density. To estimate the parameter, this method only requires $\nabla_{\boldsymbol{x}} \log p(\boldsymbol{x}; \boldsymbol{\theta})$ and $\nabla_{\boldsymbol{x}}^2 \log p(\boldsymbol{x}; \boldsymbol{\theta})$ to avoid evaluating the normalization term.

**Kernel Stein Discrepancy (KSD) [2]** is a kernel mean discrepancy measure between a data distribution and a model density using the Stein identity defined on Stein operator $T'_{p\boldsymbol{\theta}}$. This measure has been used for model evaluation [4, 19]. In Section 7, we minimize such a discrepancy with respect to $\boldsymbol{\theta}$ for unnormalized model parameter estimation. A more generic version of this estimator has been discussed in [2].

**Noise Contrastive Estimation (NCE) [10]** estimates the parameters of an unnormalized statistical model by performing a non-linear logistic regression to discriminate between observed dataset and artificially generated noise. The normalization term can be dealt with like a regular parameter in the statistical model and estimated through such a logistic regression. NCE requires us to select a noise distribution and in our experiments, we use a multivariate Gaussian distribution with mean and variance the same as $X_q$.

## 7 Experiments

### 7.1 Validation of Asymptotic Results

To examine the asymptotic distribution of $\sqrt{n_q}[\hat{\boldsymbol{\theta}} - \boldsymbol{\theta}^*]$, we design an intractable exponential family model $\bar{p}(\boldsymbol{x}; \boldsymbol{\theta}) := \exp\left[\boldsymbol{\eta}(\boldsymbol{\theta})^\top \boldsymbol{\psi}(\boldsymbol{x})\right]$, where

$$\boldsymbol{\psi}(\boldsymbol{x}) := [\sum_{i=1}^{d} x_i^2, x_1 x_2, \sum_{i=3}^{d} x_1 x_i, \tanh(\boldsymbol{x})]^\top, \boldsymbol{\eta}(\boldsymbol{\theta}) := [-.5, .6, .2, 0, 0, 0, \boldsymbol{\theta}]^\top, \boldsymbol{x} \in \mathbb{R}^5, \boldsymbol{\theta} \in \mathbb{R}^2.$$

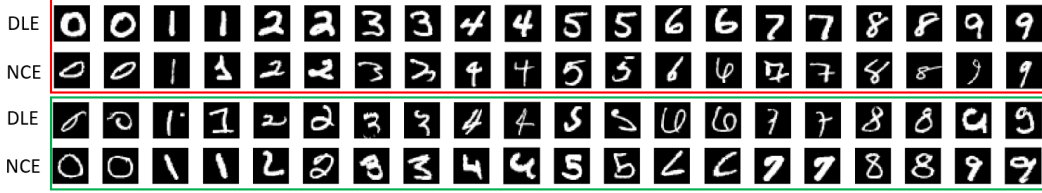

Figure 2: MNIST images with the highest (upper red box) and the lowest unnormalized density (lower green box) estimated on each digit by DLE and NCE.

$\tanh(\boldsymbol{x})$ is applied in an element-wise fashion. The feature function of the Stein feature is chosen as $\boldsymbol{f}(\boldsymbol{x}) := \tanh(\boldsymbol{x}) \in \mathbb{R}^5$. Due to the $\tanh$ function, $\bar{p}(\boldsymbol{x}; \boldsymbol{\theta})$ does not have a closed form normalization term. We draw $n_q = 500$ samples from $p(\boldsymbol{x}; \boldsymbol{0})$ as $X_q$. Given we set $\boldsymbol{\theta}^* = \boldsymbol{0}$, $X_q$ actually comes from a tractable distribution. However the intractability of $\bar{p}(\boldsymbol{x}; \boldsymbol{\theta})$ does not allow us to perform MLE straight away.

We run DLE 6000 times with new batch of $X_q$ each time and obtain an empirical distribution of $\sqrt{n_q}\hat{\boldsymbol{\theta}}$. We show qqplots of its marginal distributions vs. $\mathcal{N}(0, V_{1,1}), \mathcal{N}(0, V_{2,2})$, the asymptotic distribution predicted by Theorem 2 whose variance $V$ is approximated by $X_q$ and $\hat{\boldsymbol{\theta}}$. Figure 1a shows all quantiles between the empirical marginals and predicted marginals are very well aligned. We also scatter-plot $\sqrt{n_q}\hat{\boldsymbol{\theta}}$ together with the predicted 95% and 99.9% confidence interval in Figure 1b. It can be seen that the empirical joint distribution of $\sqrt{n_q}\hat{\boldsymbol{\theta}}$ has the same elongated shape as predicted by Theorem 2 and agrees with the predicted confidence intervals nicely.

One of our major contributions is proving DLE attains the Cramér-Rao bound. We now compare the variances of the estimated parameter $\hat{\theta}$ using Gamma $p(x; \theta) = \Gamma(5, \theta), \theta^* = 1$ and Gaussian mixture model $p(x; \theta) = .5\mathcal{N}(\theta, 1) + .5\mathcal{N}(1, 1), \theta^* = -1$ across DLE, SM and KSD. $\text{Var}_{n_q}[\hat{\theta}]$ are shown on Figure 1c and 1d. For DLE, we set $\boldsymbol{f}(\boldsymbol{x}) := [x, x^2]$ and for KSD, we use a polynomial kernel with degree 2. Note we particularly choose $p(x; \theta)$ to be tractable so we can compute MLE and Cramér-Rao bound easily. It can be seen that all estimators have decreasing variances and MLE, being one of the minimum variance estimators, has the lowest variance. However, DLE has the second lowest variances in both cases and quickly converges to Cramér-Rao bound after $n_q = 150$. In comparison, both KSD and SM maintain higher levels of variances.

### 7.2 Unnormalized Model Using Pre-trained Deep Neural Network (DNN)

In this experiment, we create an exponential family model $\bar{p}(\boldsymbol{x}; \boldsymbol{\theta}_i) := \exp[\boldsymbol{\theta}_i^\top \boldsymbol{\psi}(\boldsymbol{x})], \boldsymbol{x} \in \mathbb{R}^{784}$ where $\boldsymbol{\psi}(\boldsymbol{x}) \in \mathbb{R}^{20}$ is a pre-trained 3-layer DNN. $\boldsymbol{\psi}(\boldsymbol{x})$ is trained using a logistic regression so that the classification error is minimized on the full MNIST dataset over all digits. Clearly, $\bar{p}(\boldsymbol{x}; \boldsymbol{\theta}_i)$ does not have a tractable normalization term. The dataset $X_{q_i}$ contains $n_q = 100$ randomly selected images from a *single digit* i and we use DLE and NCE to estimate $\hat{\theta}_i$ for each digit i. For DLE, we set $\boldsymbol{f}(\boldsymbol{x}) = \boldsymbol{\psi}(\boldsymbol{x})$. Though we can only obtain an unnormalized density for each digit, it can be used to rank images and find potential inliers and outliers.

In Figure 2 we show images that are ranked either among the top two or bottom two places when sorted by $\log \bar{p}(\boldsymbol{x}; \hat{\boldsymbol{\theta}}_i)$, for each digit i. It can be seen that, when $\hat{\boldsymbol{\theta}}$ is estimated by DLE, images ranked the highest are indeed typical looking images, while the lowest ranking images tend to be outliers in that digit group. However, in comparison, when $\hat{\boldsymbol{\theta}}$ is estimated by NCE, some highest ranked images are distorted while some lowest ranked image look very regular. This experiment shows the usefulness of DLE as a model inference method when working with a complex model (DNN) on a high dimensional dataset ($d = 784$) using relatively small number of samples ($n_q = 100$).

## 8 Conclusion and Discussion

In this paper, we introduce a model inference method for unnormalized statistical models. First, Stein density ratio estimation is used to fit a ratio and to approximate the KL divergence. The model inference is done by minimizing such an approximated KL divergence. Despite promising theoretical and experimental results, future works are needed to demonstrate a systematic way of choosing Stein features in different scenarios as the performance of DLE depends heavily on such choices.

## Acknowledgements

This work was supported by The Alan Turing Institute under the EPSRC grant EP/N510129/1. TK was partially supported by JSPS KAKENHI Grant Number 15H01678, 15H03636, 16K00044, and 19H04071. Authors would like to thank Dr. Carl Henrik Ek and three anonymous reviewers for their insightful comments.

## Footnotes

[1]The word "discriminator" is borrowed from GAN [7]. Indeed, DLE and GAN bears many resemblances.

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
