[Supplementary Material]

# Fisher Efficient Inference of Intractable Models

## Supplementary

## A    Examples

### A.1    Examples of Stein Features $T_{\boldsymbol{\theta}}\boldsymbol{f}(\boldsymbol{x})$

**Example 3.** *Let $p = \mathcal{N}(0,1)$, $T_{\mathcal{N}(0,1)}1 = 0$, then $T_{\mathcal{N}(0,1)}x = -x$ and $T_{\mathcal{N}(0,1)}x^2/2 = -x^2 + 1$.*

As we see, Stein features with respect to $\mathcal{N}(0,1)$ using monomials of $x$ are same-order polynomial terms of $x$ which have been widely used as function basis in various function fitting applications.

### A.2    Assumption 2 Examples

**Example 4.** *When $f(x) = 0$, by the definition of Stein feature at Section 3.1, $T_{\boldsymbol{\theta}}f(x) \equiv 0$. Our density ratio model does not have any discriminative power and become a constant function $1$. We can see $H_{\delta,\delta} = 0$, $H_{\delta,\theta} = 0$ regardless what $\delta$ and $\theta$ are chosen. Thus, Assumption 2 is not satisfied here. See (14) and (15) in Section B.2 in Appendix for the exact formulas of $H_{\delta,\delta}$ and $H_{\delta,\theta}$.*

**Example 5.** *When $f(x) := x$ and $p(x;\theta) := \mathcal{N}(\theta,1)$, our density ratio model becomes a linear discriminative function (See Example 3). From (14) and (15) we can see, when $\theta = \theta^*$ and $\delta = 0$, $H^*_{\delta,\delta} = -\frac{1}{n_q}\sum_{i=1}^{n_q}(x_q^{(i)} - \theta^*)^2$ which is essentially the negative sample variance and $H^*_{\delta,\theta} = \frac{1}{n_q}\sum_{i=1}^{n_q}\nabla_{\theta}(x_q^{(i)} - \theta) = -1$. Given $n_q$ is sufficiently large, $\Lambda_{\min}$ and $\Lambda'_{\min}$ is reasonably small and $\Lambda_{\max}$ is reasonably large, Assumption 2 should hold at the optimal point $(\theta^*, 0)$ with high probability. We omit the analysis when $\delta$ and $\theta$ are slightly deviated from their optimal values due to the page limit. Nonetheless, it can be analysed with some extra regularity conditions.*

### A.3    Example of Asymptotic Efficient Choice of $\boldsymbol{f}(\boldsymbol{x})$

**Example 6.** *Consider the univariate Gaussian distribution $p(x;\boldsymbol{\theta}) = \exp\left\{\theta_1 x + \theta_2 x^2\right\}/Z(\boldsymbol{\theta})$ for $x \in \mathbb{R}$, $\boldsymbol{\theta} = (\theta_1, \theta_2)$, where $\theta_1 \in \mathbb{R}$, $\theta_2 < 0$, and $Z(\boldsymbol{\theta})$ is the normalization constant. The score function is $s_1(x;\boldsymbol{\theta}) = x - \frac{1}{Z(\boldsymbol{\theta})}\partial_{\theta_1}Z(\boldsymbol{\theta})$, $s_2(x;\boldsymbol{\theta}) = x^2 - \frac{1}{Z(\boldsymbol{\theta})}\partial_{\theta_2}Z(\boldsymbol{\theta})$. Let us consider the Stein feature vector for $\boldsymbol{f}(x) = (x, x^2/2)^{\top}$, $T_{\boldsymbol{\theta}}\boldsymbol{f}(\boldsymbol{x}) = (\theta_1 + 2\theta_2 x, 1 + \theta_1 x + 2\theta_2 x^2)^{\top}$. We know that $\theta_1 Z(\boldsymbol{\theta}) + 2\theta_2\partial_{\theta_1}Z(\boldsymbol{\theta}) = 0$ and $Z(\boldsymbol{\theta}) + \theta_1\partial_{\theta_1}Z(\boldsymbol{\theta}) + 2\theta_2\partial_{\theta_2}Z(\boldsymbol{\theta}) = 0$ (see [11] for details). Thus, $\begin{pmatrix} T_{\boldsymbol{\theta}}f_1(\boldsymbol{x}) \\ T_{\boldsymbol{\theta}}f_2(\boldsymbol{x}) \end{pmatrix} = \begin{pmatrix} 2\theta_2 & 0 \\ \theta_1 & 2\theta_2 \end{pmatrix}\begin{pmatrix} s_1 \\ s_2 \end{pmatrix}$. The coefficient matrix is invertible as long as $\theta_2 \neq 0$. Hence, the DLE with the above $\boldsymbol{f}$ achieves the asymptotic efficiency bound.*

## B    Proofs

For simplicity, we write all $\sum_{i=1}^{n_q} g(\boldsymbol{x}_q^{(i)})$ as $\sum_{i=1}^{n_q} g(\boldsymbol{x}^{(i)})$ from now on as samples always come from dataset $X_q$. See Table 1 for all defined notations.

### B.1    Proof of Lemma 1

*Proof.* Our proof below is similar to the proof of Lemma 4 in [13]. It can be seen that

$$\mathbb{E}_{p_{\boldsymbol{\theta}}}[T_{\boldsymbol{\theta}}f_i(\boldsymbol{x})] = \int p(\boldsymbol{x};\boldsymbol{\theta})\left[\langle \nabla_{\boldsymbol{x}}\log p(\boldsymbol{x};\boldsymbol{\theta}), \nabla_{\boldsymbol{x}}f_i(\boldsymbol{x})\rangle + \mathrm{trace}(\nabla_{\boldsymbol{x}}^2 f_i(\boldsymbol{x}))\right]\mathrm{d}\boldsymbol{x}$$

$$= \int \langle \nabla_{\boldsymbol{x}}p(\boldsymbol{x};\boldsymbol{\theta}), \nabla_{\boldsymbol{x}}f_i(\boldsymbol{x})\rangle + p(\boldsymbol{x};\boldsymbol{\theta})\cdot\mathrm{trace}(\nabla_{\boldsymbol{x}}^2 f_i(\boldsymbol{x}))\mathrm{d}\boldsymbol{x}.$$

Table 1: Notations of Symbols

| Symbol | Definition |
|---|---|
| $\ell(\boldsymbol{\delta}, \boldsymbol{\theta})$ | $\frac{1}{n_q} \sum_{i=1}^{n_q} \log r_{\boldsymbol{\theta}}(\boldsymbol{x}_q^{(i)}; \boldsymbol{\delta})$, log likelihood ratio |
| $\nabla \ell(\boldsymbol{\delta}_0, \boldsymbol{\theta}_0)$ | $\nabla_{(\boldsymbol{\delta}, \boldsymbol{\theta})} \ell(\boldsymbol{\delta}_0, \boldsymbol{\theta}_0)\|_{\boldsymbol{\delta}=\boldsymbol{\delta}_0, \boldsymbol{\theta}=\boldsymbol{\theta}_0}$ |
| $\nabla_{\boldsymbol{\delta}} \ell(\boldsymbol{\delta}_0, \boldsymbol{\theta}_0)$ | $\nabla_{\boldsymbol{\delta}} \ell(\boldsymbol{\delta}_0, \boldsymbol{\theta}_0)$ |
| $\boldsymbol{H}$ | $\nabla_{(\boldsymbol{\delta}, \boldsymbol{\theta})}^2 \ell(\boldsymbol{\delta}, \boldsymbol{\theta})$, Hessian of likelihood |
| $\boldsymbol{H}_{\boldsymbol{\delta}, \boldsymbol{\theta}}$ | $\nabla_{\boldsymbol{\delta}} \nabla_{\boldsymbol{\theta}} \ell(\boldsymbol{\delta}, \boldsymbol{\theta})$, submatrix of Hessian. |
| $\text{Ball}(R, \boldsymbol{x}_0)$ | $\ell_2$ ball with radius $R$ centered at $\boldsymbol{x}_0$ |
| $\|A\|$ | $\ell_2$ norm of a vector $A$ or the **spectral norm** of a matrix $A$ |
| $\boldsymbol{s}(\boldsymbol{x}; \boldsymbol{\theta}) \in \mathbb{R}^{\dim(\boldsymbol{\theta})}$ | $\nabla_{\boldsymbol{\theta}} \log p(\boldsymbol{x}, \boldsymbol{\theta})$, Score function of $p_{\boldsymbol{\theta}}$ |
| $\boldsymbol{s}$ | $s(\boldsymbol{x}; \boldsymbol{\theta}^*)$ |

Let us rewrite $\mathbb{E}_{p_{\boldsymbol{\theta}}}[T_{\boldsymbol{\theta}} f_i(\boldsymbol{x})]$ as nested integrals over each component of $\boldsymbol{x}$:

$$\mathbb{E}_{p_{\boldsymbol{\theta}}}[T_{\boldsymbol{\theta}} f_i(\boldsymbol{x})]$$

$$= \sum_{j=1}^{d} \int_{\boldsymbol{x}_{\backslash j}} \int_{x_j} \partial_{x_j} f_i(\boldsymbol{x}) \cdot \partial_{x_j} p(\boldsymbol{x}; \boldsymbol{\theta}) + p(\boldsymbol{x}; \boldsymbol{\theta}) \cdot \partial_{x_j}^2 f_i(\boldsymbol{x}) \mathrm{d}x_j \mathrm{d}\boldsymbol{x}_{\backslash j}, \tag{11}$$

$$= \sum_{j=1}^{d} \int_{\boldsymbol{x}_{\backslash j}} \underbrace{\left[ p(\boldsymbol{x}; \boldsymbol{\theta}) \partial_{x_j} f_i(\boldsymbol{x}) \right]_{x_j \to -\infty}^{x_j \to +\infty}}_{0, \text{by assumption}} \mathrm{d}\boldsymbol{x}_{\backslash j} - \int_{\boldsymbol{x}_{\backslash j}} \int_{x_j} p(\boldsymbol{x}; \boldsymbol{\theta}) \left[ \partial_{x_j}^2 f_i(\boldsymbol{x}) - \partial_{x_j}^2 f_i(\boldsymbol{x}) \right] \mathrm{d}x_j \mathrm{d}\boldsymbol{x}_{\backslash j} \tag{12}$$

$$= 0. \tag{13}$$

where $\boldsymbol{x}_{\backslash j}$ contains all the components in $\boldsymbol{x}$ except the $j$-th component. The equality from (11) to (12) is due to one dimensional integration by parts formula. The first term in (12) is zero as the product of $p(\boldsymbol{x})$ and $\partial_{x_j} f_i(\boldsymbol{x})$ is asssumed to be zero when $x_j$ takes the limit to $+/-\infty$. Our assumption holds for all $i, j$, so we can assert $\forall_i \mathbb{E}_{p_{\boldsymbol{\theta}}}[T_{\boldsymbol{\theta}} f_i(\boldsymbol{x})] = 0$ and $\mathbb{E}_{p_{\boldsymbol{\theta}}}[T_{\boldsymbol{\theta}} \boldsymbol{f}(\boldsymbol{x})] = \boldsymbol{0}$ by its construction. □

## B.2 Derivations of $\nabla_{\delta}^2 \ell(\delta, \boldsymbol{\theta})$ and $\nabla_{\delta, \boldsymbol{\theta}} \ell(\delta, \boldsymbol{\theta})$ with $f(\boldsymbol{x}) : \mathbb{R}^d \to \mathbb{R}$

$$\nabla_{\delta}^2 \ell(\delta, \boldsymbol{\theta}) = -\frac{1}{n_q} \sum_{i=1}^{n_q} \frac{\left[ T_{\boldsymbol{\theta}} f(\boldsymbol{x}^{(i)}) \right]^2}{r_{\boldsymbol{\theta}}^2(\boldsymbol{x}^{(i)}; \delta)} + 0, \tag{14}$$

$$\nabla_{\delta, \boldsymbol{\theta}} \ell(\delta, \boldsymbol{\theta}) = -\frac{1}{n_q} \sum_{i=1}^{n_q} \frac{T_{\boldsymbol{\theta}} f(\boldsymbol{x}^{(i)})}{r_{\boldsymbol{\theta}}^2(\boldsymbol{x}^{(i)}; \delta)} \nabla_{\boldsymbol{\theta}} r_{\boldsymbol{\theta}}(\boldsymbol{x}^{(i)}; \delta) + \frac{1}{n_q} \sum_{i=1}^{n_q} \frac{1}{r_{\boldsymbol{\theta}}(\boldsymbol{x}^{(i)}; \delta)} \nabla_{\boldsymbol{\theta}} T_{\boldsymbol{\theta}} f(\boldsymbol{x}^{(i)}). \tag{15}$$

## B.3 Proof of Proposition 2

*Proof.* First, the definition of $\Delta_{n_q}$ gives the boundedness of our ratio, i.e., $\frac{1}{C_{\text{ratio}}} \leq r_{\boldsymbol{\theta}}(\boldsymbol{x}; \boldsymbol{\delta}) \leq C_{\text{ratio}}, \forall \boldsymbol{x} \in X_q, \forall \boldsymbol{\theta} \in \Theta$.

Second, $-\boldsymbol{H}_{\boldsymbol{\delta}, \boldsymbol{\delta}} = \frac{1}{n_q} \sum_{i=1}^{n_q} \frac{1}{r_{\boldsymbol{\theta}}^2(\boldsymbol{x}^{(i)}; \boldsymbol{\delta})} \cdot T\boldsymbol{\psi}^{(i)} T\boldsymbol{\psi}^{(i)^\top}$, where $T\boldsymbol{\psi}^{(i)}$ is an abbreviation of $T_{\boldsymbol{\theta}} \boldsymbol{\psi}(\boldsymbol{x}^{(i)})$. It is a sum over ratio weighted positive semi-definite matrices so we can lower bound its minimum eigenvalue using the lower bound of the ratio:

$$\lambda_{\min}(-\boldsymbol{H}_{\boldsymbol{\delta}, \boldsymbol{\delta}}) \geq \frac{1}{C_{\text{ratio}}^2} \lambda_{\min} \left( \frac{1}{n_q} \sum_{i=1}^{n_q} T\boldsymbol{\psi}^{(i)} T\boldsymbol{\psi}^{(i)^\top} \right) > \frac{\Lambda_{\min}''}{C_{\text{ratio}}^2} > 0, \text{with high prob.},$$

due to our assumption. Similarly, we can also upper-bound its maximum eigenvalue

$$\lambda_{\max}(-\boldsymbol{H}_{\boldsymbol{\delta}, \boldsymbol{\delta}}) \leq C_{\text{ratio}}^2 \lambda_{\max} \left( \frac{1}{n_q} \sum_{i=1}^{n_q} T\boldsymbol{\psi}^{(i)} T\boldsymbol{\psi}^{(i)^\top} \right) \leq C_{\text{ratio}}^2 \Lambda_{\max}'', \text{with high prob.},$$

Third, $-\boldsymbol{H}_{\boldsymbol{\theta},\boldsymbol{\theta}} = \frac{1}{n_q} \sum_{i=1}^{n_q} \frac{1}{r_{\boldsymbol{\theta}}^2(\boldsymbol{x}^{(i)};\boldsymbol{\delta})} \boldsymbol{J}_{\boldsymbol{x}}\boldsymbol{\psi}(\boldsymbol{x}^{(i)})\boldsymbol{J}_{\boldsymbol{x}}\boldsymbol{\psi}(\boldsymbol{x}^{(i)})^{\top}\boldsymbol{\delta}\boldsymbol{\delta}^{\top}\boldsymbol{J}_{\boldsymbol{x}}\boldsymbol{\psi}(\boldsymbol{x}^{(i)})^{\top}\boldsymbol{J}_{\boldsymbol{x}}\boldsymbol{\psi}(\boldsymbol{x}^{(i)})$. We can see

$$\|\boldsymbol{H}_{\boldsymbol{\theta},\boldsymbol{\theta}}\| \leq \frac{C_{\text{ratio}}^2 \cdot \|\boldsymbol{\delta}\|^2}{n_q} \sum_{i=1}^{n_q} \|\boldsymbol{J}_{\boldsymbol{x}}\boldsymbol{\psi}(\boldsymbol{x}^{(i)})\|^4 \leq C_{\text{ratio}}^2 C_2 \cdot \|\boldsymbol{\delta}\|^2 \leq \frac{C_{\text{ratio}}^2 C_2 T}{\sigma(n_q)^2}.$$

Fourth, using the fact that $-\boldsymbol{H}_{\boldsymbol{\delta},\boldsymbol{\delta}}$ is a positive definite matrix, which we have just proved, we can see

$$
\begin{aligned}
\lambda_{\min}\left\{-\boldsymbol{H}_{\boldsymbol{\theta},\boldsymbol{\delta}}\boldsymbol{H}_{\boldsymbol{\delta},\boldsymbol{\delta}}^{-1}\boldsymbol{H}_{\boldsymbol{\delta},\boldsymbol{\theta}}\right\} &= \lambda_{\min}(-\boldsymbol{H}_{\boldsymbol{\delta},\boldsymbol{\delta}}^{-1}\boldsymbol{H}_{\boldsymbol{\delta},\boldsymbol{\theta}}\boldsymbol{H}_{\boldsymbol{\theta},\boldsymbol{\delta}}) \\
&\geq \lambda_{\min}(-\boldsymbol{H}_{\boldsymbol{\delta},\boldsymbol{\delta}}^{-1})\lambda_{\min}(\boldsymbol{H}_{\boldsymbol{\delta},\boldsymbol{\theta}}\boldsymbol{H}_{\boldsymbol{\theta},\boldsymbol{\delta}}) \\
&= \frac{\lambda_{\min}(\boldsymbol{H}_{\boldsymbol{\delta},\boldsymbol{\theta}}\boldsymbol{H}_{\boldsymbol{\theta},\boldsymbol{\delta}})}{\lambda_{\max}(-\boldsymbol{H}_{\boldsymbol{\delta},\boldsymbol{\delta}})} \geq \frac{\lambda_{\min}(\boldsymbol{H}_{\boldsymbol{\delta},\boldsymbol{\theta}}\boldsymbol{H}_{\boldsymbol{\theta},\boldsymbol{\delta}})}{C_{\text{ratio}}^2 \Lambda_{\max}''},
\end{aligned}
$$

where 2nd line is due to Theorem 7, [21]. So we only need to find a lower bound for $\lambda_{\min}(\boldsymbol{H}_{\boldsymbol{\delta},\boldsymbol{\theta}}\boldsymbol{H}_{\boldsymbol{\theta},\boldsymbol{\delta}})$. We can write $\boldsymbol{H}_{\boldsymbol{\theta},\boldsymbol{\delta}}$ as

$$\boldsymbol{H}_{\boldsymbol{\theta},\boldsymbol{\delta}} = \underbrace{\frac{1}{n_q} \sum_{i=1}^{n_q} \frac{1}{r_{\boldsymbol{\theta}}(\boldsymbol{x}^{(i)};\boldsymbol{\delta})} \boldsymbol{J}_{\boldsymbol{x}}\boldsymbol{\psi}(\boldsymbol{x}^{(i)})\boldsymbol{J}_{\boldsymbol{x}}\boldsymbol{\psi}(\boldsymbol{x}^{(i)})^{\top}}_{A} \tag{16}$$

$$\underbrace{-\frac{1}{n_q} \sum_{i=1}^{n_q} \frac{1}{r_{\boldsymbol{\theta}}^2(\boldsymbol{x}^{(i)};\boldsymbol{\delta})} \boldsymbol{J}_{\boldsymbol{x}}\boldsymbol{\psi}(\boldsymbol{x}^{(i)})\boldsymbol{J}_{\boldsymbol{x}}\boldsymbol{\psi}(\boldsymbol{x}^{(i)})^{\top}\boldsymbol{\delta}T_{\boldsymbol{\theta}}\boldsymbol{\psi}(\boldsymbol{x}^{(i)})^{\top}}_{B} \tag{17}$$

Therefore $\boldsymbol{H}_{\boldsymbol{\delta},\boldsymbol{\theta}}\boldsymbol{H}_{\boldsymbol{\theta},\boldsymbol{\delta}}$ can be written as

$$\boldsymbol{A}\boldsymbol{A}^{\top} - \boldsymbol{A}\boldsymbol{B}^{\top} - \boldsymbol{B}\boldsymbol{A}^{\top} + \boldsymbol{B}\boldsymbol{B}^{\top}.$$

Since we are analyzing the minimum eigenvalue, we can safely ignore the last term $\boldsymbol{B}\boldsymbol{B}^{\top}$ as it is positive semi-definite. This gives the following inequality:

$$
\begin{aligned}
\lambda_{\min}\left\{\boldsymbol{A}\boldsymbol{A}^{\top} - \boldsymbol{A}\boldsymbol{B}^{\top} - \boldsymbol{B}\boldsymbol{A}^{\top}\right\} &\geq \lambda_{\min}\left\{\boldsymbol{A}\boldsymbol{A}^{\top}\right\} + \lambda_{\min}\left\{-\boldsymbol{A}\boldsymbol{B}^{\top} - \boldsymbol{B}\boldsymbol{A}^{\top}\right\} \\
&\geq \lambda_{\min}\left\{\boldsymbol{A}\boldsymbol{A}^{\top}\right\} - \|\boldsymbol{A}\boldsymbol{B}^{\top} + \boldsymbol{B}\boldsymbol{A}^{\top}\|
\end{aligned}
$$

As $\boldsymbol{A}$ is a sum of ratio weighted positive semi-definite matrices, we can use the same trick in the second step to lower bound its eigenvalue using the lower bound of the density ratio, eventually, using our assumption on $\lambda_{\min}\{\frac{1}{n_q} \sum_{i=1}^{n_q} \boldsymbol{J}^{(i)}\boldsymbol{J}^{(i)\top}\} \geq C_3$, we can get,

$$\lambda_{\min}(\boldsymbol{A}) \geq \frac{C_3}{C_{\text{ratio}}}, \lambda_{\min}(\boldsymbol{A}\boldsymbol{A}^{\top}) \geq \lambda_{\min}(\boldsymbol{A}) \cdot \lambda_{\min}(\boldsymbol{A}) \geq \frac{C_3^2}{C_{\text{ratio}}^2}.$$

Now we analyze $\|\boldsymbol{A}\boldsymbol{B}^{\top} + \boldsymbol{B}\boldsymbol{A}^{\top}\|$ which is further upperbounded by $2\|\boldsymbol{A}\|\|\boldsymbol{B}\|$.

Similarly to how $\lambda_{\min}(\boldsymbol{A})$ is bounded, we can upper-bound $\|\boldsymbol{A}\|$ using the upperbound of the ratio: $\|\boldsymbol{A}\| \leq C_{\text{ratio}}C_4$. Let us write $\boldsymbol{B} = \frac{1}{n_q} \sum_{i=1}^{n_q} \frac{1}{r_i^2}\boldsymbol{J}^{(i)}\boldsymbol{J}^{(i)\top}\boldsymbol{\delta}T\boldsymbol{\psi}^{(i)\top}$ where $\boldsymbol{J}^{(i)}$ and $r_i$ are abbreviations of $\boldsymbol{J}_{\boldsymbol{x}}\boldsymbol{\psi}(\boldsymbol{x}^{(i)})$ and $r_{\boldsymbol{\theta}}(\boldsymbol{x}^{(i)};\boldsymbol{\delta})$. It can be seen that

$$
\begin{aligned}
\|\boldsymbol{B}\| &\leq \frac{1}{n_q} \sum_{i=1}^{n_q} \|\frac{1}{r_i^2}\boldsymbol{J}^{(i)}\boldsymbol{J}^{(i)\top}\| \cdot \|\boldsymbol{\delta}T\boldsymbol{\psi}^{(i)}\| \leq \frac{1}{n_q} \sum_{i=1}^{n_q} \|\frac{1}{r_i^2}\boldsymbol{J}^{(i)}\boldsymbol{J}^{(i)\top}\| \cdot \|\boldsymbol{\delta}\| \cdot \|T\boldsymbol{\psi}^{(i)}\|, \\
&\leq C_{\text{ratio}}^2 \cdot C_5 T/\sigma(n_q).
\end{aligned}
$$

Now we can bound

$$
\begin{aligned}
\lambda_{\min}\left\{\boldsymbol{A}\boldsymbol{A}^{\top} - \boldsymbol{A}\boldsymbol{B}^{\top} - \boldsymbol{B}\boldsymbol{A}^{\top} + \boldsymbol{B}\boldsymbol{B}^{\top}\right\} &\geq \lambda_{\min}\left\{\boldsymbol{A}\boldsymbol{A}^{\top}\right\} - 2\|\boldsymbol{A}\|\|\boldsymbol{B}\| \\
&\geq \frac{C_3^2}{C_{\text{ratio}}^2} - C_{\text{ratio}}^3 C_4 \cdot C_5 T/\sigma(n_q)
\end{aligned}
$$

There exists a constant $N > 0$, such that when $n_q > N$,

$$\lambda_{\min}\left\{-\boldsymbol{H}_{\boldsymbol{\theta},\boldsymbol{\delta}}\boldsymbol{H}_{\boldsymbol{\delta},\boldsymbol{\delta}}^{-1}\boldsymbol{H}_{\boldsymbol{\delta},\boldsymbol{\theta}}\right\} \geq \frac{\lambda_{\min}(\boldsymbol{H}_{\boldsymbol{\delta},\boldsymbol{\theta}}\boldsymbol{H}_{\boldsymbol{\theta},\boldsymbol{\delta}})}{C_{\text{ratio}}^2\Lambda_{\max}''} \geq \frac{C_3^2}{C_{\text{ratio}}^4\Lambda_{\max}'} - \frac{C_{\text{ratio}}C_4 \cdot C_5 T}{\sigma(n_q)\Lambda_{\max}''}$$

$$\geq \frac{C_{\text{ratio}}^2 C_2 T}{\sigma(n_q)^2} \geq \|\boldsymbol{H}_{\boldsymbol{\theta},\boldsymbol{\theta}}\|.$$

Finally we analyze $\left\|\boldsymbol{H}_{\boldsymbol{\theta},\boldsymbol{\delta}}\boldsymbol{H}_{\boldsymbol{\delta},\boldsymbol{\delta}}^{-1}\right\| \cdot \left\|\boldsymbol{H}_{\boldsymbol{\theta},\boldsymbol{\delta}}\boldsymbol{H}_{\boldsymbol{\delta},\boldsymbol{\delta}}^{-1}\right\| \leq \|\boldsymbol{H}_{\boldsymbol{\theta},\boldsymbol{\delta}}\| \cdot \left\|\boldsymbol{H}_{\boldsymbol{\delta},\boldsymbol{\delta}}^{-1}\right\|$. As $-\boldsymbol{H}_{\boldsymbol{\delta},\boldsymbol{\delta}}$ is positive definite, the operator norm of its inverse is the inverse of its minimum eigenvalue, which is upperbounded by $C_{\text{ratio}}^2/\Lambda_{\min}''$. On the other hand, we can rewrite (16) as $\boldsymbol{H}_{\boldsymbol{\theta},\boldsymbol{\delta}} = \frac{1}{n_q}\sum_{i=1}^{n_q}\frac{1}{r_i} \cdot \boldsymbol{J}^{(i)}\boldsymbol{J}^{(i)^\top} \cdot \left(\text{Iden} - \frac{1}{r_i} \cdot \boldsymbol{\delta}T\boldsymbol{\psi}^{(i)^\top}\right)$, so

$$\|\boldsymbol{H}_{\boldsymbol{\theta},\boldsymbol{\delta}}\| \leq \frac{1}{n_q}\sum_{i=1}^{n_q}\frac{1}{r_i} \cdot \left\|\boldsymbol{J}^{(i)}\boldsymbol{J}^{(i)^\top} \cdot \left(\text{Iden} - \frac{1}{r_i} \cdot \boldsymbol{\delta}T\boldsymbol{\psi}^{(i)^\top}\right)\right\|$$

$$\leq \frac{C_{\text{ratio}}}{n_q}\sum_{i=1}^{n_q}\|\boldsymbol{J}^{(i)}\boldsymbol{J}^{(i)^\top}\| \cdot \underbrace{\|\text{Iden} - \frac{1}{r_i} \cdot \boldsymbol{\delta}T\boldsymbol{\psi}^{(i)^\top}\|}_{C}$$

From calculation, we know $\|\boldsymbol{C}\| \leq 1 + |(r_i - 1)/r_i| \leq 2 + C_{\text{ratio}}$. Therefore $\|\boldsymbol{H}_{\boldsymbol{\theta},\boldsymbol{\delta}}\| \leq \frac{C_{\text{ratio}}^2 + 2C_{\text{ratio}}}{n_q}\sum_{i=1}^{n_q}\|\boldsymbol{J}^{(i)}\boldsymbol{J}^{(i)^\top}\| \leq (C_{\text{ratio}}^2 + 2C_{\text{ratio}})C_4$. Therefore $\left\|\boldsymbol{H}_{\boldsymbol{\theta},\boldsymbol{\delta}}\boldsymbol{H}_{\boldsymbol{\delta},\boldsymbol{\delta}}^{-1}\right\|$ is upperbounded by $(C_{\text{ratio}}^4 + 2C_{\text{ratio}}^3)C_4/\Lambda_{\min}''$.

Refer to [21, 6] for inequalities of eigenvalue of matrix summation and product. $\qquad\square$

## B.4  Proof of Theorem 1

*Proof.* We denote Hessian $\boldsymbol{H}$ as a block matrix:

$$\boldsymbol{H} = \nabla^2\ell(\boldsymbol{\delta},\boldsymbol{\theta}) = \begin{pmatrix} \boldsymbol{H}_{11} & \boldsymbol{H}_{12} \\ \boldsymbol{H}_{21} & \boldsymbol{H}_{22} \end{pmatrix} = \begin{pmatrix} \nabla_{\boldsymbol{\delta}}^2\ell(\boldsymbol{\delta},\boldsymbol{\theta}) & \nabla_{\boldsymbol{\delta}}\nabla_{\boldsymbol{\theta}}\ell(\boldsymbol{\delta},\boldsymbol{\theta}) \\ \nabla_{\boldsymbol{\theta}}\nabla_{\boldsymbol{\delta}}\ell(\boldsymbol{\delta},\boldsymbol{\theta}) & \nabla_{\boldsymbol{\theta}}^2\ell(\boldsymbol{\delta},\boldsymbol{\theta}) \end{pmatrix},$$

then Assumption 2 states that for every $\boldsymbol{\delta} \in \Delta_{n_q}$ and $\boldsymbol{\theta} \in \Theta$, $\lambda(\boldsymbol{H}_{21}\boldsymbol{H}_{11}^{-1}\boldsymbol{H}_{12})$ is lower bounded by $2\|\boldsymbol{H}_{22}\|$ and $\|\boldsymbol{H}_{21}\boldsymbol{H}_{11}^{-1}\|$ is upper bounded.

We can write the optimality condition of (7) and expand them at $(\boldsymbol{\delta}^* \equiv \boldsymbol{0}, \boldsymbol{\theta}^*)$:

$$\nabla_{\boldsymbol{\delta}}\ell(\hat{\boldsymbol{\delta}},\hat{\boldsymbol{\theta}}) = \boldsymbol{0} = \nabla_{\boldsymbol{\delta}}\ell(\boldsymbol{\delta}^*,\boldsymbol{\theta}^*) + \bar{\boldsymbol{H}}_{11}(\hat{\boldsymbol{\delta}} - \boldsymbol{\delta}^*) + \bar{\boldsymbol{H}}_{12}(\hat{\boldsymbol{\theta}} - \boldsymbol{\theta}^*) \tag{18}$$

$$\nabla_{\boldsymbol{\theta}}\ell(\hat{\boldsymbol{\delta}},\hat{\boldsymbol{\theta}}) = \boldsymbol{0} = \nabla_{\boldsymbol{\theta}}\ell(\boldsymbol{\delta}^*,\boldsymbol{\theta}^*) + \bar{\boldsymbol{H}}_{21}(\hat{\boldsymbol{\delta}} - \boldsymbol{\delta}^*) + \bar{\boldsymbol{H}}_{22}(\hat{\boldsymbol{\theta}} - \boldsymbol{\theta}^*), \tag{19}$$

where $\bar{\boldsymbol{H}}$ is the Hessian evaluated at a $(\bar{\boldsymbol{\delta}},\bar{\boldsymbol{\theta}})$ which is in between $(\hat{\boldsymbol{\delta}},\hat{\boldsymbol{\theta}})$ and $(\boldsymbol{\delta}^*,\boldsymbol{\theta}^*)$ in *an element-wise fashion*. This expansion is basically one-dimensional mean-value theorem applied on *each individual dimension* of $\nabla_{\boldsymbol{\delta}}\ell(\hat{\boldsymbol{\delta}},\hat{\boldsymbol{\theta}})$ and $\nabla_{\boldsymbol{\theta}}\ell(\hat{\boldsymbol{\delta}},\hat{\boldsymbol{\theta}})$.

Given (18) and (19) we can solve equations for $\hat{\boldsymbol{\delta}} - \boldsymbol{\delta}^*$ and $\hat{\boldsymbol{\theta}} - \boldsymbol{\theta}^*$.

From (18) we can get

$$\hat{\boldsymbol{\delta}} - \boldsymbol{\delta}^* = \bar{\boldsymbol{H}}_{11}^{-1}\left[-\nabla_{\boldsymbol{\delta}}\ell(\boldsymbol{\delta}^*,\boldsymbol{\theta}^*) - \bar{\boldsymbol{H}}_{12}(\hat{\boldsymbol{\theta}} - \boldsymbol{\theta}^*)\right]. \tag{20}$$

Substituting (20) into (19) we get

$$\boldsymbol{0} = \nabla_{\boldsymbol{\theta}}\ell(\boldsymbol{\delta}^*,\boldsymbol{\theta}^*) - \bar{\boldsymbol{H}}_{21}\bar{\boldsymbol{H}}_{11}^{-1}\nabla_{\boldsymbol{\delta}}\ell(\boldsymbol{\delta}^*,\boldsymbol{\theta}^*) + \left[-\bar{\boldsymbol{H}}_{21}\bar{\boldsymbol{H}}_{11}^{-1}\bar{\boldsymbol{H}}_{12} + \bar{\boldsymbol{H}}_{22}\right]\left(\hat{\boldsymbol{\theta}} - \boldsymbol{\theta}^*\right).$$

Rearranging terms, we get

$$\hat{\boldsymbol{\theta}} - \boldsymbol{\theta}^* = \left[\bar{\boldsymbol{H}}_{21}\bar{\boldsymbol{H}}_{11}^{-1}\bar{\boldsymbol{H}}_{12} - \bar{\boldsymbol{H}}_{22}\right]^{-1}\left(\nabla_{\boldsymbol{\theta}}\ell(\boldsymbol{\delta}^*,\boldsymbol{\theta}^*) - \bar{\boldsymbol{H}}_{21}\bar{\boldsymbol{H}}_{11}^{-1}\nabla_{\boldsymbol{\delta}}\ell(\boldsymbol{\delta}^*,\boldsymbol{\theta}^*)\right) \tag{21}$$

$$= \left[-\bar{\boldsymbol{H}}_{21}\bar{\boldsymbol{H}}_{11}^{-1}\bar{\boldsymbol{H}}_{12} + \bar{\boldsymbol{H}}_{22}\right]^{-1}\bar{\boldsymbol{H}}_{21}\bar{\boldsymbol{H}}_{11}^{-1}\nabla_{\boldsymbol{\delta}}\ell(\boldsymbol{\delta}^*,\boldsymbol{\theta}^*). \tag{22}$$

The last line uses the fact that $\nabla_{\boldsymbol{\theta}}\ell(\boldsymbol{\delta}^*, \boldsymbol{\theta}^*) \equiv \mathbf{0}$.

Weyl's inequality states:

$$\lambda_{\min}(A + B) \geq \lambda_{\min}(A) + \lambda_{\min}(B).$$

As $\bar{\boldsymbol{\delta}} \in \Delta_{n_q}$ and $\bar{\boldsymbol{\theta}} \in \Theta$, $\bar{\boldsymbol{H}}$ is regulated by Assumption 2. Since

$$\lambda_{\min}(-\bar{\boldsymbol{H}}_{21}\bar{\boldsymbol{H}}_{11}^{-1}\bar{\boldsymbol{H}}_{12}) \geq \Lambda_{\min}$$

and

$$\lambda_{\min}(\bar{\boldsymbol{H}}_{22}) \geq -\|\bar{\boldsymbol{H}}_{22}\| \geq -\frac{\Lambda_{\min}}{2}$$

which are assumed by Assumption 2, we have

$$\lambda_{\min}(-\bar{\boldsymbol{H}}_{21}\bar{\boldsymbol{H}}_{11}^{-1}\bar{\boldsymbol{H}}_{12} + \bar{\boldsymbol{H}}_{22}) \geq \Lambda_{\min}/2 > 0.$$

Denote $-\bar{\boldsymbol{H}}_{21}\bar{\boldsymbol{H}}_{11}^{-1}\bar{\boldsymbol{H}}_{12} + \bar{\boldsymbol{H}}_{22}$ as $\bar{\boldsymbol{H}}/\bar{\boldsymbol{H}}_{22}$ (it is actually the Schur Complement of $\bar{\boldsymbol{H}}$). Using Holder's inequality, we get

$$\|\hat{\boldsymbol{\theta}} - \boldsymbol{\theta}^*\| \leq \left\|\left[\bar{\boldsymbol{H}}/\bar{\boldsymbol{H}}_{22}\right]^{-1}\right\| \left\|\bar{\boldsymbol{H}}_{21}\bar{\boldsymbol{H}}_{11}^{-1}\right\| \|\nabla_{\boldsymbol{\delta}}\ell(\boldsymbol{\delta}^*, \boldsymbol{\theta}^*)\|$$

$$\leq \frac{\left\|\bar{\boldsymbol{H}}_{21}\bar{\boldsymbol{H}}_{11}^{-1}\right\|}{\lambda_{\min}\left[\bar{\boldsymbol{H}}/\bar{\boldsymbol{H}}_{22}\right]} \cdot \|\nabla_{\boldsymbol{\delta}}\ell(\boldsymbol{\delta}^*, \boldsymbol{\theta}^*)\| \leq \frac{2\Lambda_{\max}}{\Lambda_{\min}} \cdot \|\nabla_{\boldsymbol{\delta}}\ell(\boldsymbol{\delta}^*, \boldsymbol{\theta}^*)\|. \quad (23)$$

Further, we have $\mathbb{E}_q[T_{\boldsymbol{\theta}^*}\boldsymbol{f}(\boldsymbol{x})] = \mathbb{E}_{p_{\boldsymbol{\theta}^*}}[T_{\boldsymbol{\theta}^*}\boldsymbol{f}(\boldsymbol{x})] = \mathbf{0}$. The first equality is due to Assumption 1 and the second equality is given by Stein identity.

Therefore, $\nabla_{\boldsymbol{\delta}}\ell(\boldsymbol{\delta}^*, \boldsymbol{\theta}^*) = \frac{1}{n_q}\sum_{i=1}^{n_q} T_{\boldsymbol{\theta}^*}f(\boldsymbol{x}^{(i)}) - \mathbf{0} = \frac{1}{n_q}\sum_{i=1}^{n_q} T_{\boldsymbol{\theta}^*}f(\boldsymbol{x}^{(i)}) - \mathbb{E}_q[T_{\boldsymbol{\theta}^*}\boldsymbol{f}(\boldsymbol{x})]$, which converges to 0 in $\ell_2$ norm in probability due to Assumption 3. This gives the convergence in probability of $\|\hat{\boldsymbol{\theta}} - \boldsymbol{\theta}^*\|$. Finite sample convergence rate can be given if the convergence rate of $\|\nabla_{\boldsymbol{\delta}}\ell(\boldsymbol{\delta}^*, \boldsymbol{\theta}^*)\|$ is known.

Now we show the consistency of $\hat{\boldsymbol{\delta}}$. From (20) we can see that

$$\hat{\boldsymbol{\delta}} - \boldsymbol{\delta}^* = -\bar{\boldsymbol{H}}_{11}^{-1}\nabla_{\boldsymbol{\delta}}\ell(\boldsymbol{\delta}^*, \boldsymbol{\theta}^*) - \bar{\boldsymbol{H}}_{11}^{-1}\bar{\boldsymbol{H}}_{12}(\hat{\boldsymbol{\theta}} - \boldsymbol{\theta}^*),$$

and due to Holder's inequality, we get

$$\left\|\hat{\boldsymbol{\delta}} - \boldsymbol{\delta}^*\right\| = \left\|-\bar{\boldsymbol{H}}_{11}^{-1}\right\| \|\nabla_{\boldsymbol{\delta}}\ell(\boldsymbol{\delta}^*, \boldsymbol{\theta}^*)\| + \left\|\bar{\boldsymbol{H}}_{11}^{-1}\bar{\boldsymbol{H}}_{12}\right\| \left\|\hat{\boldsymbol{\theta}} - \boldsymbol{\theta}^*\right\|$$

$$\leq \frac{1}{\Lambda'_{\min}}\|\nabla_{\boldsymbol{\delta}}\ell(\boldsymbol{\delta}^*, \boldsymbol{\theta}^*)\| + \Lambda_{\max}\left\|\hat{\boldsymbol{\theta}} - \boldsymbol{\theta}^*\right\|. \quad (24)$$

Combine (24) with (23) we get

$$\left\|\hat{\boldsymbol{\delta}} - \boldsymbol{\delta}^*\right\| \leq \frac{2\Lambda_{\max}^2\Lambda'_{\min} + \Lambda_{\min}}{\Lambda_{\min}\Lambda'_{\min}} \cdot \|\nabla_{\boldsymbol{\delta}}\ell(\boldsymbol{\delta}^*, \boldsymbol{\theta}^*)\|$$

Again, due to Assumption 3, $\|\nabla_{\boldsymbol{\delta}}\ell(\boldsymbol{\delta}^*, \boldsymbol{\theta}^*)\| \xrightarrow{\mathbb{P}} \mathbf{0}$. This completes the proof.

$\square$

## B.5 Proof of Theorem 2

*Proof.* Due to Assumption 4, it can be seen that $\bar{\boldsymbol{H}} \xrightarrow{\mathbb{P}} \mathbb{E}_q[\bar{\boldsymbol{H}}]$. Moreover, as $\bar{\boldsymbol{\theta}} \xrightarrow{\mathbb{P}} \boldsymbol{\theta}^*$ and $\bar{\boldsymbol{\delta}} \xrightarrow{\mathbb{P}} \mathbf{0}$ (proved in Theorem 1), we can see $\mathbb{E}_q[\bar{\boldsymbol{H}}] \xrightarrow{\mathbb{P}} \mathbb{E}_q[\boldsymbol{H}^*]$ due to continuous mapping. Thus $\bar{\boldsymbol{H}} = \mathbb{E}_q[\boldsymbol{H}^*] + o_p(1)$. From now on, for simplicity, let us denote $-\mathbb{E}_q[\boldsymbol{H}^*]$ as $\boldsymbol{I}$ [2].

We again write the optimality condition of (7) and apply asymptotic expansion at $(\boldsymbol{\delta}^* \equiv \mathbf{0}, \boldsymbol{\theta}^*)$:

$$\nabla_{\boldsymbol{\delta}}\ell(\hat{\boldsymbol{\delta}}, \hat{\boldsymbol{\theta}}) = \mathbf{0} = \nabla_{\boldsymbol{\delta}}\ell(\boldsymbol{\delta}^*, \boldsymbol{\theta}^*) + (-\boldsymbol{I}_{11} + o_p(1))(\hat{\boldsymbol{\delta}} - \boldsymbol{\delta}^*) + (-\boldsymbol{I}_{12} + o_p(1))(\hat{\boldsymbol{\theta}} - \boldsymbol{\theta}^*) \quad (25)$$

$$\nabla_{\boldsymbol{\theta}}\ell(\hat{\boldsymbol{\delta}}, \hat{\boldsymbol{\theta}}) = \mathbf{0} = \nabla_{\boldsymbol{\theta}}\ell(\boldsymbol{\delta}^*, \boldsymbol{\theta}^*) + (-\boldsymbol{I}_{21} + o_p(1))(\hat{\boldsymbol{\delta}} - \boldsymbol{\delta}^*) + (-\boldsymbol{I}_{22} + o_p(1))(\hat{\boldsymbol{\theta}} - \boldsymbol{\theta}^*). \quad (26)$$

Note we have replaced all $\bar{\boldsymbol{H}}$ with $-\boldsymbol{I} + o_p(1)$, and $o_p(1)$ will be ignored in future algebraic calculations.

We now get an asymptotic version of (22):

$$\sqrt{n_q}\left(\hat{\boldsymbol{\theta}} - \boldsymbol{\theta}^*\right) \rightsquigarrow -\left(\boldsymbol{I}_{21}\boldsymbol{I}_{11}^{-1}\boldsymbol{I}_{12} - \boldsymbol{I}_{22}\right)^{-1} \boldsymbol{I}_{21}\boldsymbol{I}_{11}^{-1}\nabla_{\boldsymbol{\delta}}\ell(\boldsymbol{\delta}^*, \boldsymbol{\theta}^*) \cdot \sqrt{n_q}$$

$$= -\left(\boldsymbol{I}_{21}\boldsymbol{I}_{11}^{-1}\boldsymbol{I}_{12}\right)^{-1} \boldsymbol{I}_{21}\boldsymbol{I}_{11}^{-1}\nabla_{\boldsymbol{\delta}}\ell(\boldsymbol{\delta}^*, \boldsymbol{\theta}^*) \cdot \sqrt{n_q}$$

The last equality is due to $\boldsymbol{I}_{22} \equiv \mathbf{0}$.

Noticing that $\boldsymbol{I}_{11}^{-1}\nabla_{\boldsymbol{\delta}}\ell(\boldsymbol{\delta}^*, \boldsymbol{\theta}^*) \cdot \sqrt{n_q}$ is a sum of independent random variables with zero mean and covariance $-\boldsymbol{I}_{11}^{-1}$. Applying CLT on $\boldsymbol{I}_{11}^{-1}\nabla_{\boldsymbol{\delta}}\ell(\boldsymbol{\delta}^*, \boldsymbol{\theta}^*) \cdot \sqrt{n_q}$ yields

$$\boldsymbol{I}_{11}^{-1}\nabla_{\boldsymbol{\delta}}\ell(\boldsymbol{\delta}^*, \boldsymbol{\theta}^*) \rightsquigarrow \mathcal{N}\left(\mathbf{0}, -\boldsymbol{I}_{11}^{-1}\right),$$

thus

$$\sqrt{n_q}\left(\hat{\boldsymbol{\theta}} - \boldsymbol{\theta}^*\right) \rightsquigarrow \mathcal{N}\left[\mathbf{0}, \left(-\boldsymbol{I}_{21}\boldsymbol{I}_{11}^{-1}\boldsymbol{I}_{12}\right)^{-1}\right].$$

$\square$

## B.6  Proof of Lemma 3

*Proof.* Let us shorten the Stein feature vector $T_{\boldsymbol{\theta}}\boldsymbol{f}(\boldsymbol{x})$ as $\boldsymbol{t}(\boldsymbol{x}; \boldsymbol{\theta}) \in \mathbb{R}^b$ and $\boldsymbol{t}$ as $\boldsymbol{t}(\boldsymbol{x}; \boldsymbol{\theta}^*)$. We start by computing each factors in the variance. Since $r_{\boldsymbol{\theta}}(\boldsymbol{x}; \boldsymbol{\delta}^*) = 1$ holds for all $\boldsymbol{\theta}$, we have $\nabla_{\boldsymbol{\theta}}r_{\boldsymbol{\theta}}(\boldsymbol{x}; \boldsymbol{\delta}^*) = \mathbf{0}$. Then, we have

$$-\mathbb{E}_q\left[\boldsymbol{H}_{\boldsymbol{\delta},\boldsymbol{\delta}}^*\right] = -\mathbb{E}_q\left[\nabla_{\boldsymbol{\delta}}^2 \log r_{\boldsymbol{\theta}^*}(\boldsymbol{x}; \boldsymbol{\delta}^*)\right]$$

$$= \mathbb{E}_q\left[\frac{1}{r(\boldsymbol{x}; \boldsymbol{\delta}^*, \boldsymbol{\theta}^*)^2}\boldsymbol{t}\boldsymbol{t}^\top\right] = \mathbb{E}_q[\boldsymbol{t}\boldsymbol{t}^\top] \in \mathbb{R}^{b \times b},$$

$$\mathbb{E}_q[\boldsymbol{H}_{\boldsymbol{\theta},\boldsymbol{\delta}}^*] = \mathbb{E}_q\left[\frac{1}{r}\nabla_{\boldsymbol{\theta}}\boldsymbol{t}(\boldsymbol{x}; \boldsymbol{\theta}^*)^\top - \frac{1}{r^2}\nabla_{\boldsymbol{\theta}}r_{\boldsymbol{\theta}^*}(\boldsymbol{x}; \boldsymbol{\delta}^*)\boldsymbol{t}(\boldsymbol{x}; \boldsymbol{\theta}^*)^\top\right]$$

$$= \mathbb{E}_q\left[\nabla_{\boldsymbol{\theta}}\boldsymbol{t}(\boldsymbol{x}; \boldsymbol{\theta}^*)^\top\right] \in \mathbb{R}^{\dim(\boldsymbol{\theta}) \times b}.$$

Since the equality $\mathbb{E}_{p_{\boldsymbol{\theta}}}[\boldsymbol{t}(\boldsymbol{x}; \boldsymbol{\theta})] = \mathbf{0}$ holds for all $\boldsymbol{\theta}$, we have $\nabla_{\boldsymbol{\theta}}\mathbb{E}_{p_{\boldsymbol{\theta}}}[\boldsymbol{t}(\boldsymbol{x}; \boldsymbol{\theta})] = \mathbf{0}$. Exchangeability of the integration and the derivative yields

$$\nabla_{\boldsymbol{\theta}}\mathbb{E}_{p_{\boldsymbol{\theta}}}[\boldsymbol{t}(\boldsymbol{x}; \boldsymbol{\theta})] = \mathbb{E}_{p_{\boldsymbol{\theta}}}\left[\boldsymbol{s}(\boldsymbol{x}; \boldsymbol{\theta})\boldsymbol{t}(\boldsymbol{x}; \boldsymbol{\theta})^\top\right] + \mathbb{E}_{p_{\boldsymbol{\theta}}}\left[\nabla_{\boldsymbol{\theta}}\boldsymbol{t}(\boldsymbol{x}; \boldsymbol{\theta})^\top\right] = \mathbf{0}.$$

As a result, we obtain

$$\mathbb{E}_q[\boldsymbol{H}_{\boldsymbol{\theta},\boldsymbol{\delta}}^*] = -\mathbb{E}_q[\boldsymbol{s}\boldsymbol{t}^\top].$$

$\square$

## B.7  Proof of Theorem 5

*Proof.* Use Taylor series to expand $\mathbb{E}_q\left[\ell(\hat{\boldsymbol{\delta}}, \hat{\boldsymbol{\theta}})\right]$ on $(\boldsymbol{\theta}^*, \boldsymbol{\delta}^*)$, we get

$$\mathbb{E}_q\left[\ell(\hat{\boldsymbol{\delta}}, \hat{\boldsymbol{\theta}})\right] = \mathbb{E}_q[\ell(\boldsymbol{\delta}^*, \boldsymbol{\theta}^*)] + \nabla_{\boldsymbol{\delta}}\mathbb{E}_q[\ell(\boldsymbol{\delta}^*, \boldsymbol{\theta}^*)]^\top\left[\hat{\boldsymbol{\delta}} - \boldsymbol{\delta}^*\right] + \nabla_{\boldsymbol{\theta}}\mathbb{E}_q[\ell(\boldsymbol{\delta}^*, \boldsymbol{\theta}^*)]^\top\left[\hat{\boldsymbol{\theta}} - \boldsymbol{\theta}^*\right]$$

$$+ \frac{1}{2}[\hat{\boldsymbol{\eta}} - \boldsymbol{\eta}^*]^\top \nabla_{\boldsymbol{\eta}}^2\mathbb{E}_q[\ell(\bar{\boldsymbol{\eta}})][\hat{\boldsymbol{\eta}} - \boldsymbol{\eta}^*]$$

$$= 0 + 0 + 0 + \frac{1}{2}[\hat{\boldsymbol{\eta}} - \boldsymbol{\eta}^*]^\top \nabla_{\boldsymbol{\eta}}^2\mathbb{E}_q[\ell(\bar{\boldsymbol{\eta}})][\hat{\boldsymbol{\eta}} - \boldsymbol{\eta}^*] \quad (27)$$

where we denote $\boldsymbol{\eta} := \begin{bmatrix} \boldsymbol{\delta} \\ \boldsymbol{\theta} \end{bmatrix}$ for short and $\bar{\boldsymbol{\eta}}$ is defined in between $\hat{\boldsymbol{\eta}}$ and $\boldsymbol{\eta}^*$ in an element-wise fashion.

The second equality is due to $\boldsymbol{\delta}^* = \mathbf{0}$ and $\mathbb{E}_q\left[\nabla_{\boldsymbol{\delta}}\ell(\boldsymbol{\delta}^*, \boldsymbol{\theta}^*)\right] = \mathbf{0}$, which is given by Stein identity. Similarly we can expand

$$\ell(\hat{\boldsymbol{\delta}}, \hat{\boldsymbol{\theta}}) = \nabla_{\boldsymbol{\delta}}\ell(\boldsymbol{\delta}^*, \boldsymbol{\theta}^*)^\top \left[\hat{\boldsymbol{\delta}} - \boldsymbol{\delta}^*\right] + \frac{1}{2}\left[\hat{\boldsymbol{\eta}} - \boldsymbol{\eta}^*\right]^\top \nabla_{\boldsymbol{\eta}}^2 \ell(\bar{\boldsymbol{\eta}})\left[\hat{\boldsymbol{\eta}} - \boldsymbol{\eta}^*\right], \tag{28}$$

where $\bar{\bar{\boldsymbol{\eta}}}$ is similarly defined as $\bar{\boldsymbol{\eta}}$. It can be seen that $\nabla_{\boldsymbol{\eta}}^2 \ell(\bar{\bar{\boldsymbol{\eta}}}) \xrightarrow{\mathbb{P}} -\boldsymbol{I}$ and $\nabla_{\boldsymbol{\eta}}^2 \mathbb{E}_q\left[\ell\bar{\boldsymbol{\eta}}\right] \xrightarrow{\mathbb{P}} -\boldsymbol{I}$ due to Assumption 4 and our consistency results. Taking the difference between (27) and (28) after multiplying $n_q$ yields

$$n_q \mathbb{E}_q\left[\ell(\hat{\boldsymbol{\delta}}, \hat{\boldsymbol{\theta}})\right] - n_q \ell(\hat{\boldsymbol{\delta}}, \hat{\boldsymbol{\theta}}) = -n_q \nabla_{\boldsymbol{\delta}}\ell(\boldsymbol{\delta}^*, \boldsymbol{\theta}^*)^\top \left[\hat{\boldsymbol{\delta}} - \boldsymbol{\delta}^*\right] + o_p(1).$$

Substitute $(\hat{\boldsymbol{\delta}} - \boldsymbol{\delta}^*)$ with (20) we get

$$n_q \mathbb{E}_q\left[\ell(\hat{\boldsymbol{\delta}}, \hat{\boldsymbol{\theta}})\right] - n_q \ell(\hat{\boldsymbol{\delta}}, \hat{\boldsymbol{\theta}}) = n_q \nabla_{\boldsymbol{\delta}}\ell(\boldsymbol{\delta}^*, \boldsymbol{\theta}^*)^\top \left[\bar{\boldsymbol{H}}_{11}^{-1}\nabla_{\boldsymbol{\delta}}\ell(\boldsymbol{\delta}^*, \boldsymbol{\theta}^*) + \bar{\boldsymbol{H}}_{11}^{-1}\bar{\boldsymbol{H}}_{12}(\hat{\boldsymbol{\theta}} - \boldsymbol{\theta}^*)\right] + o_p(1).$$

Substitute $(\hat{\boldsymbol{\theta}} - \boldsymbol{\theta}^*)$ using (22), we get

$$n_q \mathbb{E}_q\left[\ell(\hat{\boldsymbol{\delta}}, \hat{\boldsymbol{\theta}})\right] - n_q \ell(\hat{\boldsymbol{\delta}}, \hat{\boldsymbol{\theta}}) = n_q \nabla_{\boldsymbol{\delta}}\ell(\boldsymbol{\delta}^*, \boldsymbol{\theta}^*)^\top \bar{\boldsymbol{H}}_{11}^{-1}\nabla_{\boldsymbol{\delta}}\ell(\boldsymbol{\delta}^*, \boldsymbol{\theta}^*)$$
$$- n_q \nabla_{\boldsymbol{\delta}}\ell(\boldsymbol{\delta}^*, \boldsymbol{\theta}^*)^\top \bar{\boldsymbol{H}}_{11}^{-1}\bar{\boldsymbol{H}}_{12}\left[\bar{\boldsymbol{H}}/\bar{\boldsymbol{H}}_{22}\right]^{-1}\bar{\boldsymbol{H}}_{21}\bar{\boldsymbol{H}}_{11}^{-1}\nabla_{\boldsymbol{\delta}}\ell(\boldsymbol{\delta}^*, \boldsymbol{\theta}^*) + o_p(1) \tag{29}$$

Replacing submatrices of $\bar{\boldsymbol{H}}_{a,b}$ using submatrices of $-\boldsymbol{I}_{a,b}$ in (29) and using the fact that $\boldsymbol{I}_{22} \equiv \mathbf{0}$ (due to $\boldsymbol{\delta}^* = \mathbf{0}$),

$$n_q \mathbb{E}_q\left[\ell(\hat{\boldsymbol{\delta}}, \hat{\boldsymbol{\theta}})\right] - n_q \ell(\hat{\boldsymbol{\delta}}, \hat{\boldsymbol{\theta}}) = -\sqrt{n_q}\nabla_{\boldsymbol{\delta}}\ell(\boldsymbol{\delta}^*, \boldsymbol{\theta}^*)^\top \boldsymbol{I}_{11}^{-1}\nabla_{\boldsymbol{\delta}}\ell(\boldsymbol{\delta}^*, \boldsymbol{\theta}^*)\sqrt{n_q}$$
$$+ \sqrt{n_q}\nabla_{\boldsymbol{\delta}}\ell(\boldsymbol{\delta}^*, \boldsymbol{\theta}^*)^\top \boldsymbol{I}_{11}^{-1}\boldsymbol{I}_{12}\left[\boldsymbol{I}_{21}\boldsymbol{I}_{11}^{-1}\boldsymbol{I}_{12}\right]^{-1}\boldsymbol{I}_{21}\boldsymbol{I}_{11}^{-1}\nabla_{\boldsymbol{\delta}}\ell(\boldsymbol{\delta}^*, \boldsymbol{\theta}^*)\sqrt{n_q} + o_p(1) \tag{30}$$

Taking the expectation,

$$n_q \mathbb{E}\left\{\mathbb{E}_q\left[\ell(\hat{\boldsymbol{\delta}}, \hat{\boldsymbol{\theta}})\right] - \ell(\hat{\boldsymbol{\delta}}, \hat{\boldsymbol{\theta}})\right\} = -\operatorname{trace}(\boldsymbol{I}_{11}\boldsymbol{I}_{11}^{-1}) + \operatorname{trace}(\boldsymbol{I}_{11}^{-1}\boldsymbol{I}_{12}\left[\boldsymbol{I}_{21}\boldsymbol{I}_{11}^{-1}\boldsymbol{I}_{12}\right]^{-1}\boldsymbol{I}_{21}) + o_p(1)$$
$$= -\operatorname{rank}(\boldsymbol{I}_{11}) + \operatorname{rank}\left(\boldsymbol{I}_{21}\boldsymbol{I}_{11}^{-1}\boldsymbol{I}_{12}\right) + o_p(1).$$

In the case when $\boldsymbol{I}_{11} \in \mathbb{R}^{b \times b}, \boldsymbol{I}_{12} \in \mathbb{R}^{b \times \dim(\boldsymbol{\theta})}$ are full-rank and $\dim(\boldsymbol{\theta}) \leq b$, $\operatorname{rank}(\boldsymbol{I}_{11}) = b$ and $\operatorname{rank}\left(\boldsymbol{I}_{21}\boldsymbol{I}_{11}^{-1}\boldsymbol{I}_{12}\right) = \dim(\boldsymbol{\theta})$, which completes the proof. $\qquad\square$

## B.8 The Asymptotic Distribution of $2n_q \ell(\hat{\boldsymbol{\delta}}, \hat{\boldsymbol{\theta}})$

We show $2n_q \ell(\hat{\boldsymbol{\delta}}, \hat{\boldsymbol{\theta}})$ follows a $\chi^2$ distribution based on previously assumed assumptions.

**Theorem 6.** *Suppose Assumption 1, 2, 3 and 4 holds, $\mathbb{E}_q\left[\boldsymbol{H}_{\boldsymbol{\delta}, \boldsymbol{\delta}}^*\right]$ is invertible and $\mathbb{E}_q\left[\boldsymbol{H}_{\boldsymbol{\delta}, \boldsymbol{\theta}}^*\right]$ are full-rank and $\dim(\boldsymbol{\theta}) < b$, then $2n_q \ell(\hat{\boldsymbol{\delta}}, \hat{\boldsymbol{\theta}}) \rightsquigarrow \chi^2(b - \dim(\boldsymbol{\theta}))$.*

*Proof.* First we expand $2n_q \ell(\hat{\boldsymbol{\delta}}, \hat{\boldsymbol{\theta}})$ using mean value theorem:

$$2n_q \ell(\hat{\boldsymbol{\delta}}, \hat{\boldsymbol{\theta}}) = 2n_q \nabla_{\boldsymbol{\delta}}\ell(\boldsymbol{\delta}^*, \boldsymbol{\theta}^*)^\top d\boldsymbol{\delta} + n_q d\boldsymbol{\delta}\bar{\boldsymbol{H}}_{11}d\boldsymbol{\delta} + n_q d\boldsymbol{\delta}\bar{\boldsymbol{H}}_{12}d\boldsymbol{\theta} + n_q d\boldsymbol{\theta}\bar{\boldsymbol{H}}_{21}d\boldsymbol{\delta} + n_q d\boldsymbol{\theta}\bar{\boldsymbol{H}}_{22}d\boldsymbol{\theta} \tag{31}$$

where $d\boldsymbol{t}$ is short for $\hat{\boldsymbol{t}} - \boldsymbol{t}^*$. Note $\ell(\boldsymbol{\delta}^*, \boldsymbol{\theta}^*) = 0$. Now we analyze each term.

From the proof in Section B.7 we know

$$2n_q \nabla_{\boldsymbol{\delta}}\ell(\boldsymbol{\delta}^*, \boldsymbol{\theta}^*)^\top d\boldsymbol{\delta} = 2n_q \nabla_{\boldsymbol{\delta}}\ell(\boldsymbol{\delta}^*, \boldsymbol{\theta}^*)^\top \boldsymbol{I}_{11}^{-1}\nabla_{\boldsymbol{\delta}}\ell(\boldsymbol{\delta}^*, \boldsymbol{\theta}^*)$$
$$- 2n_q \nabla_{\boldsymbol{\delta}}\ell(\boldsymbol{\delta}^*, \boldsymbol{\theta}^*)^\top \boldsymbol{I}_{11}^{-1}\boldsymbol{I}_{12}\left[\boldsymbol{I}_{21}\boldsymbol{I}_{11}^{-1}\boldsymbol{I}_{12}\right]^{-1}\boldsymbol{I}_{21}\boldsymbol{I}_{11}^{-1}\nabla_{\boldsymbol{\delta}}\ell(\boldsymbol{\delta}^*, \boldsymbol{\theta}^*) + o_p(1). \tag{32}$$

With the help of (20) and (22) and a few algebra we can see that

$$n_q d\boldsymbol{\delta} \boldsymbol{I}_{11} d\boldsymbol{\delta} = n_q \nabla_{\boldsymbol{\delta}} \ell(\boldsymbol{\delta}^*, \boldsymbol{\theta}^*)^\top \boldsymbol{I}_{11}^{-1} \nabla_{\boldsymbol{\delta}} \ell(\boldsymbol{\delta}^*, \boldsymbol{\theta}^*)$$
$$- n_q \nabla_{\boldsymbol{\delta}} \ell(\boldsymbol{\delta}^*, \boldsymbol{\theta}^*)^\top \boldsymbol{I}_{11}^{-1} \boldsymbol{I}_{12} \left[ \boldsymbol{I}_{21} \boldsymbol{I}_{11}^{-1} \boldsymbol{I}_{12} \right]^{-1} \boldsymbol{I}_{21} \boldsymbol{I}_{11}^{-1} \nabla_{\boldsymbol{\delta}} \ell(\boldsymbol{\delta}^*, \boldsymbol{\theta}^*) + o_p(1). \quad (33)$$

Similar calculations also show $n_q d\boldsymbol{\delta} \bar{\boldsymbol{H}}_{12} d\boldsymbol{\theta} = n_q d\boldsymbol{\theta} \bar{\boldsymbol{H}}_{21} d\boldsymbol{\delta} = o_p(1)$ and $n_q d\boldsymbol{\theta} \bar{\boldsymbol{H}}_{22} d\boldsymbol{\theta} = o_p(1)$. Combine (31), (32) and (33), we can see that

$$2n_q \ell(\hat{\boldsymbol{\delta}}, \hat{\boldsymbol{\theta}}) = n_q \nabla_{\boldsymbol{\delta}} \ell(\boldsymbol{\delta}^*, \boldsymbol{\theta}^*)^\top \boldsymbol{I}_{11}^{-1} \nabla_{\boldsymbol{\delta}} \ell(\boldsymbol{\delta}^*, \boldsymbol{\theta}^*)$$
$$- n_q \nabla_{\boldsymbol{\delta}} \ell(\boldsymbol{\delta}^*, \boldsymbol{\theta}^*)^\top \boldsymbol{I}_{11}^{-1} \boldsymbol{I}_{12} \left[ \boldsymbol{I}_{21} \boldsymbol{I}_{11}^{-1} \boldsymbol{I}_{12} \right]^{-1} \boldsymbol{I}_{21} \boldsymbol{I}_{11}^{-1} \nabla_{\boldsymbol{\delta}} \ell(\boldsymbol{\delta}^*, \boldsymbol{\theta}^*) + o_p(1)$$
$$= \sqrt{n_q} \nabla_{\boldsymbol{\delta}} \ell(\boldsymbol{\delta}^*, \boldsymbol{\theta}^*)^\top \boldsymbol{I}_{11}^{-1} \left\{ \text{Iden} - \boldsymbol{I}_{12} \left[ \boldsymbol{I}_{21} \boldsymbol{I}_{11}^{-1} \boldsymbol{I}_{12} \right]^{-1} \boldsymbol{I}_{21} \boldsymbol{I}_{11}^{-1} \right\} \nabla_{\boldsymbol{\delta}} \ell(\boldsymbol{\delta}^*, \boldsymbol{\theta}^*) \sqrt{n_q} + o_p(1),$$

where Iden is identify matrix. Denote $\text{Iden} - \boldsymbol{I}_{12} \left[ \boldsymbol{I}_{21} \boldsymbol{I}_{11}^{-1} \boldsymbol{I}_{12} \right]^{-1} \boldsymbol{I}_{21} \boldsymbol{I}_{11}^{-1}$ as $A$. One can verify that $\nabla_{\boldsymbol{\delta}} \ell(\boldsymbol{\delta}^*, \boldsymbol{\theta}^*)^\top \boldsymbol{I}_{11}^{-1} A$ has covariance $\boldsymbol{I}_{11}^{-1} A$ [3]. By checking the eigenvalues of $A$ [4], it can be seen that $\text{rank}(A) = db - \dim(\boldsymbol{\theta})$ and assuming $\boldsymbol{I}_{11}^{-1}$ is full rank, $\text{rank}(\boldsymbol{I}_{11}^{-1} A) = db - \dim(\boldsymbol{\theta})$. Therefore $\sqrt{n_q} \nabla_{\boldsymbol{\delta}} \ell(\boldsymbol{\delta}^*, \boldsymbol{\theta}^*)^\top \boldsymbol{I}_{11}^{-1} A$ is asymptotically a degenerated multivariate normal variable with covariance matrix $\boldsymbol{I}_{11}^{-1} A$.

We can rewrite $\sqrt{n_q} \nabla_{\boldsymbol{\delta}} \ell(\boldsymbol{\delta}^*, \boldsymbol{\theta}^*)^\top \boldsymbol{I}_{11}^{-1} A \nabla_{\boldsymbol{\delta}} \ell(\boldsymbol{\delta}^*, \boldsymbol{\theta}^*) \sqrt{n_q}$ as

$$\sqrt{n_q} \nabla_{\boldsymbol{\delta}} \ell(\boldsymbol{\delta}^*, \boldsymbol{\theta}^*)^\top \boldsymbol{I}_{11}^{-1} A \left[ \boldsymbol{I}_{11}^{-1} A \right]^+ \boldsymbol{I}_{11}^{-1} A \nabla_{\boldsymbol{\delta}} \ell(\boldsymbol{\delta}^*, \boldsymbol{\theta}^*) \sqrt{n_q},$$

where $T^+$ is the pseudoinverse. This quadratic form has a $\chi^2$ distribution with degree of freedom $\text{rank}(\boldsymbol{I}_{11}^{-1} A) = db - \dim(\boldsymbol{\theta})$. $\qquad\square$

### B.9 Proof of Proposition 3

*Proof.* We convert the SDRE problem (5) as the following equivalent problem:

$$\max_{\boldsymbol{\delta}, \boldsymbol{\epsilon}} \sum_{i=1}^{n_q} \log \epsilon_i \quad \text{s.t. :} \forall i \in \{1 \dots n_q\}, \; \boldsymbol{\delta}^\top T_{\boldsymbol{\theta}} \boldsymbol{f}(\boldsymbol{x}_q^{(i)}) + 1 = \epsilon_i.$$

Let us introduce Lagrangian multipliers $\mu_1 \dots \mu_{n_q}$ over all the constraints. We can write the Lagrangian:

$$\min_{\boldsymbol{\mu}} \max_{\boldsymbol{\delta}, \boldsymbol{\epsilon}} \sum_{i=1}^{n_q} (\log \epsilon_i) - \sum_{i=1}^{n_q} \mu_i \left( \boldsymbol{\delta}^\top T_{\boldsymbol{\theta}} \boldsymbol{f}(\boldsymbol{x}_q^{(i)}) + 1 - \epsilon_i \right) \qquad (34)$$

Solve the inner max problem with respect to $\boldsymbol{\epsilon}$,

$$\max_{\boldsymbol{\epsilon}} \sum_{i=1}^{n_q} \log \epsilon_i + \mu_i \epsilon_i = \sum_{i=1}^{n_q} [-(\log -\mu_i) - 1], \qquad (35)$$

when $\epsilon_i = -\frac{1}{\mu_i}$. This also implies the relationship between the dual parameter $\mu_i$ and the primal parameter $\boldsymbol{\delta}$: $r_{\boldsymbol{\theta}}(\boldsymbol{x}_q^{(i)}; \boldsymbol{\delta}) = \boldsymbol{\delta}^\top T_{p_{\boldsymbol{\theta}}} \boldsymbol{f}(\boldsymbol{x}_q^{(i)}) + 1 = \epsilon_i = -\frac{1}{\mu_i}$.

The inner optimization with respect to $\boldsymbol{\delta}$, i.e., $\max_{\boldsymbol{\delta}} - \sum_{i=1}^{n_q} \mu_i \boldsymbol{\delta}^\top T_{\boldsymbol{\theta}} \boldsymbol{f}(\boldsymbol{x}_q^{(i)})$ is a linear programming and is only bounded when $\sum_{i=1}^{n_q} \mu_i T_{\boldsymbol{\theta}} \boldsymbol{f}(\boldsymbol{x}_q^{(i)}) = \boldsymbol{0}$ and achieves the optimal value 0.

Substituting the optimal values of these two maximization results into the Lagrangian and adding constraint $\sum_{i=1}^{n_q} \mu_i T_{\boldsymbol{\theta}} \boldsymbol{f}(\boldsymbol{x}_q^{(i)}) = \boldsymbol{0}$ gives the Lagrangian dual (9). Moreover, the primal problem in (5) is concave, we can verify the Slater's condition holds at $\boldsymbol{\delta} = \boldsymbol{0}, \boldsymbol{\epsilon} = \boldsymbol{1}$ thus the strong duality holds. $\qquad\square$

## Footnotes

[2] $\boldsymbol{I}$ for "information matrix". Do not confuse with the identify matrix which is denoted as $\mathrm{Iden}$ in this paper

[3]Some calculations show $A^\top \boldsymbol{I}_{11}^{-1} A = \boldsymbol{I}_{11}^{-1} A$.

[4]$\text{eig}(\text{Iden} - T) = 1 - \text{eig}(T)$ and $\text{eig}(ST) = \text{eig}(TS)$.