[Reviews · NeurIPS 2019]

Reviewer 1



The paper proposes to a KL-minimisation method based on the ratio q/p by a parametrised "ratio" model constructed from the Stein operator which automatically handles the constraint leading to a concave maximisation problem for the ratio. Consistency results are given but these assume the parameter space is compact (and other not so simple to check assumptions) which do not seem to hold in the numerical setting. The authors also show asymptotic variance converges to the inverse Fisher information, though this further assumes a (fairly strong) condition of uniform convergence which is hard to check. Nonetheless I think the contribution is quite significant as it provides a method to minimise the KL divergence without knowledge of normalising constants.

Reviewer 2



Overall, the paper has some new nice ideas for fitting a parametric model when density may only be known in an unnormalized fashion. The idea of using Stein's method to estimate the ratio function is a neat one, as it skirts the need to constrain the integral of the ratio to be 1. However, after reading the paper, I'm still unsure how much better this is than other methods people use to fit models when the normalizing constant is intractable. The experiments compared the new method to score matching and something called "KSD^2", but it didn't compare to other methods like Contrastive Divergence or Noise Contrastive Estimation. I would be more convinced if I could how much better DLE performed on more non-toy examples. Another large question I had after reading this paper was how a practitioner could select the Stein features to use. On one hand, the paper states one must be careful from selecting too many features, as then the optimization might be too pessimistic when discriminating the data from the model. On the other hand, the paper also demonstrates that the variance of the parameter estimates achieve the Rao-Cramer lower bound when the score function is in the span of the Stein features. Given these conflicting forces, how does one go about choosing the Stein features? How does one even know if the parameters being estimated are close to the optimal parameters minimizing the KL-divergence of q and p? L78: its actually the expectation of the log ratio. L98-99: Why does correlation complicate things? It doesn't appear any moment other than the first are used in the derivation of KLIEP. L179: I think the lambda is missing a subscript? Also what does it mean to hold "with high probability?" L243-250: This part is a bit confusing. Given we are trying to fit a parametric model, why would we be worried about picking up spurious patterns in the dataset? If its also a concern that we could pick too many features, how can one choose them in a principled way? This seems discordant with Corollary 4. Originality: The paper has some nice novel ideas for estimating parameters in models where the normalization constant may not be known. Using Stein's method for estimating the ratio between p and q is a neat one and worth exploring more. Quality: The paper appears to be correct. Some of the technical assumptions are quite specific and I imagine hard to check, but they do posit quantities that represent the asymptotic variance of their estimators. Clarity: The paper is well written. There are some details left out (e.g. what's KSD^2?) and some of the bounds in Assumption 2 could be written a bit more cleanly. It would also be helpful to flush out in words what these assumptions mean conceptually. Significance: The paper would be a nice addition to the set of approaches one can take for unnormalized density models. However, I'm a bit concerned on how one would apply these ideas in a general setting. APPENDUM: I would say that my original two critiques still stand. I think this paper has some interesting ideas [like using Stein's method for the density ratio estimation problem and also the asymptotic analysis accompanying their estimator] but I'm still not sure how useful this estimator is in the wild. In the author rebuttal, they say """ Contrastive divergence was commonly used for restricted Boltzman Machine (RBM) where Gibbs sampling can be efficiently done. However, we consider a much wider family of models. In our MNIST experiment, the model is more omplicated than an RBM and Gibbs sampling is hard. """ It's really not difficult to wire up an RBM to the MNIST dataset, in fact, it's a natural model to try. Hence I'm not convinced by their argument and think their paper would greatly benefit from a comparison. I've personally tried to use Stein's method for parameter estimation and found it harder to beat out other methods like NCE or Contrastive divergence. I also didn't get exactly the answer I was looking for regarding the choice of Stein features. I understand they have worked out a criteria for selecting them from a set, but I was curious how they choice the basis functions in the first place. Are they polynomials of varying degrees? Do they need to be centered near the mode of the distribution? I think there's more in the details they haven't flushed out in this paper. So overall, I'm still a weak reject, as I find the empirical results a bit lacking.

Reviewer 3



The authors present a novel algorithm with theoretical analysis and empirical results. We have a few comments and suggestions for the work: The authors note on line 139 that it is not possible to guarantee that the ratio is positive on the x domain. While the log-barrier ensures the positivity at the sample points, its non-positivity at other inputs may imply that that the estimated density p(x,\theta) is not positive everywhere. Although the positivity will hold asymptotically (if the log-barrier is active at all input points), this may be problematic in the finite-sample setting where the experiments are performed and important for the downstream use of the estimators. It would be great if the authors could comment on whether this non-positivity of r was observed in their numerical experiments and if this may be corrected in practice for high-dimensional densities? In section 4, the authors also note that the estimator resembles GANs by minimizing a discriminator. We recommend that the authors extend this comparison -- possibly in future work! For the theoretical results, it would be great if the authors could comment on how restrictive is the second assumption or its special case -- is it possible to verify these conditions hold in practice? Furthermore, in practice are the model parameters identifiable -- for instance, can another linear combination of parameters, other than \delta = 0, also ensure that r = 1? It would be great to compare assumptions 1, 2, and 3 to the assumptions imposed by SM or other consistent methods in their analysis. In the numerical results, do the authors apply the penalized likelihood that is proposed in section 4.3 for model selection? What is the numerical performance of the unpenalized vs. penalized likelihoods for the experiments in section 6? We also recommend the authors to consider proposing finite-sample likelihood estimators that minimize the out-of-sample performance to prevent overfitting (e.g., based on Leave-One-Out or other nearly unbiased estimators). We recommend the authors extend their experiment in Figure 1.c) for n > 200 to show that the DLE eventually matches the CR bound or MLE results. It also would be great if the authors could comment on how the feature functions were selected in the numerical experiments as well as the adequacy of defining the Stein features using the sufficient statistics of the exponential family distribution in Proposition 2. Lastly, we had a few minor suggestions for the text: correcting a small grammatical error on lines 85-86, including the dimensions of all defined parameters (e.g., \delta on lines 87 or 109) to help readers, clarifying the relevance of the Stein operator T' that is introduced on line 121 (i.e., could this operator also be used within a DLE context?), defining \Gamma on line 286, clarifying the models for data generation in section 6.1, and providing more details in the text for how to perform the numerical optimization in equation (10).

[Author Response · NeurIPS 2019]

We thank reviewers for their insightful comments and here we would like to address some questions raised in the review.

**R1: "Consistency results are given but these assume the parameter space is compact (and other not so simple
to check assumptions)..."** The compactness condition is only used to define a domain on which Assumption 2 and
4 hold. If Assumption 2 and 4 are defined over a neighbourhood region of the true parameter, we can remove the
compactness condition by adding an extra proof which shows $(\hat{\boldsymbol{\delta}}, \hat{\boldsymbol{\theta}})$ eventually fall in such a neighbourhood, but doing
so would introduce further technical complications. The compactness condition is among a set of conditions commonly
used in classic consistency proofs (see e.g., Wald's Consistency Proof, 5.2.1, van der Vaart, 1998). It is possible to
derive weaker conditions given specific choices of $\boldsymbol{f}$ or $p(\boldsymbol{x}; \boldsymbol{\theta})$. However, in the current manuscript, we only focus on
more *generic* settings and conditions that would give rise to estimation consistency and useful asymptotic theories.

**R1: "... though this further assumes a (fairly strong) condition of uniform convergence ..."** The uniform
convergence on Hessian is needed to control the residual of the asymptotic expansion (eq. 25, 26) and is a slight
modification of a classical regularity condition on the uniformly bounded third order derivative (5.3, van der Vaart,
1998). Again, this assumption may be weakened given specific choices of $\boldsymbol{f}$ and $p(\boldsymbol{x}; \boldsymbol{\theta})$ but we focus on investigating
generic settings where specific choices of $\boldsymbol{f}$ and $p(\boldsymbol{x}; \boldsymbol{\theta})$ are not available.

**"R1: it would be good to compare DLE to for example KSD on a
complex model. R2: more empirical examples on non-toy datasets..."**

We run the same typical/outlier image detection task in Section 6.2 on
Fashion MNIST dataset and compare DLE and KSD (see the figure). Both
methods work well and seem to assign high likelihood to similar images.
However, the tails of the fitted densities seem different, judging from low
likelihood images. We observe similar results on MNIST dataset and will
provide analysis into the differences of tail behaviors in the revision.

**"R2: ... but it didn't compare to other methods like Contrastive Diver-
gence or Noise Contrastive Estimation (NCE)."**

Contrastive divergence was commonly used for restricted Boltzman Machine (RBM) where Gibbs sampling can be
efficiently done. However, we consider a much wider family of models. In our MNIST experiment, the model is more
complicated than an RBM and Gibbs sampling is hard. Other MCMC methods such as Metropolis-Hasting are unlikely
to succeed as it is also difficult to design a proposal distribution in a 784-dimensional space. Due to the difficulties of
applying MCMC in high dimensional tasks, we restrict our discussion on sampling-free methods for their computational
efficiency and reliability in those applications. We tried NCE in the MNIST experiment but cannot find a good noise
distribution which would give comparable results to DLE and KSD. Those results will be presented in revision.

**R2:"... how a practitioner could select the Stein features to use?" R3:
"...some guidelines or heuristics for how to select the feature.."**

Section 4.3 provides an information-criterion based model selection method.
Suppose $M$ is a set of different choices of Stein features. Given a parameter
$\theta$, one should select the features $\hat{m}(\boldsymbol{\theta}) := \arg\max_{m \in M} \mathbb{E}_q[\ell(\boldsymbol{\theta}, \hat{\boldsymbol{\delta}}(m))]$ as
this choice would minimize $\mathrm{KL}[q \| r_{\hat{\boldsymbol{\delta}}} p_{\boldsymbol{\theta}}]$ (see eq. 2).

If we have a set of candidate density models $M'$, we can jointly se-
lect density model and Stein feature at the same time: $(\hat{m}, \hat{m}') :=$
$\arg\min_{m' \in M'} \max_{m \in M} \mathbb{E}_q[\ell(\hat{\boldsymbol{\theta}}(m'), \hat{\boldsymbol{\delta}}(m))]$, where $(\hat{\boldsymbol{\theta}}(m'), \hat{\boldsymbol{\delta}}(m))$ are
estimated parameters under the model choice $(m', m)$. Replacing
$\mathbb{E}_q[\ell(\boldsymbol{\theta}, \hat{\boldsymbol{\delta}}(m))]$ with the penalized likelihood derived in Section 4.3, we can
get a practical model selection method. We create a numerical experiment and plot the calculated penalized likelihood
using scaled colors with respect to both $M$ and $M'$ (see the figure on the right). It shows that our information criterion
can indeed select the optimal density model. We will explain this procedure in our revision.

**R2: "Given these conflicting forces, how does one choose the Stein features?"** Yes, there *can* be a trade-off between
efficiency and overfitting. This happens in classic settings too: MLE is an efficient estimator, but suffers from overfitting
when the dataset is small. Given a small number of samples, we may have to settle for a less efficient estimator to avoid
overfitting. However, the aforementioned information criterion can be used to select Stein features in this setting.

**R3: "...may imply that that the estimated density $p(x, \boldsymbol{\theta})$ is not positive everywhere..."** The unnormalized density
model $\bar{p}(\boldsymbol{x}; \boldsymbol{\theta})$ in our problem, by definition, should be non-negative everywhere for all $\boldsymbol{\theta} \in \boldsymbol{\Theta}$, thus the estimated
density $\bar{p}(\boldsymbol{x}; \hat{\boldsymbol{\theta}})$ is also non-negative. The estimated *density ratio* is guaranteed to be positive only within $X_q$, but the
estimated density is guaranteed to be positive everywhere by definition. We will clarify this in our revision.

[Meta-Review · NeurIPS 2019]

Two reviewers felt that this submission represents an important contribution to the field. Please be sure to carefully review and address the concerns of all reviewers in the revision.